# Climate change projected to impact structural hillslope connectivity at the global scale

Alexander T. Michalek[1], Gabriele Villarini [1,2] ✉ & Admin Husic [3]

Structural connectivity describes how landscapes facilitate the transfer of matter and plays a critical role in the flux of water, solutes, and sediment across the Earth's surface. The strength of a landscape's connectivity is a function of climatic and tectonic processes, but the importance of these drivers is poorly understood, particularly in the context of climate change. Here, we provide global estimates of structural connectivity at the hillslope level and develop a model to describe connectivity accounting for tectonic and climate processes. We find that connectivity is primarily controlled by tectonics, with climate as a second order control. However, we show climate change is projected to alter global-scale connectivity at the end of the century (2070 to 2100) by up to 4% for increasing greenhouse gas emission scenarios. Notably, the Ganges River, the world's most populated basin, is projected to experience a large increase in connectivity. Conversely, the Amazon River and the Pacific coast of Patagonia are projected to experience the largest decreases in connectivity. Modeling suggests that, as the climate warms, it could lead to increased erosion in source areas, while decreased rainfall may hinder sediment flow downstream, affecting landscape connectivity with implications for human and environmental health.

Extreme events, such as floods, droughts, and tropical storms, modulate the pathways that allow for the transport of water, nutrients, and sediment across the Earth's surface[1]. In turn, the activation of these pathways regulates the health of downstream aquatic and terrestrial ecosystems[1]. To this end, climate change is expected to exacerbate extreme events, which will alter the connectedness of hydrological and ecological pathways[2]. Areas in which droughts become more prevalent will have fragmented landscape pathways that will decrease the health of current habitats due to the reduction in nutrient supply[3]. Conversely, where extreme rainfall events become more common, damage to infrastructure and water supply will occur through flooding, sedimentation, and pollution[4–7]. To this end, an improved understanding of the degree to which climate, and consequently

climate change, controls landscape interactions is crucial to improving the resilience of global hydrologic and ecological systems[8,9].

Hillslope connectivity refers to the linkage of upstream sources and downstream transport pathways (e.g., rills, gullies, rivers) and is informed by topographic features, which themselves encode the tectonic and climatic history of a landscape[10,11]. Specifically, hillslope connectivity is composed of two interconnected types of connectivity, structural and functional[12]. Structural (static) connectivity describes the spatial arrangement of the system components, which establishes the long-term potential for downstream transport[13,14]. Functional (dynamic) connectivity is the interplay of spatial and temporal fluxes within the system for the short-term (i.e., storm event response)[15,16]. The coevolution can be expressed as the initial landscape arrangement

[1]Department of Civil and Environmental Engineering, Princeton University, Princeton, NJ, USA. [2]High Meadows Environmental Institute, Princeton University, Princeton, NJ, USA. [3]Department of Civil, Architectural, and Environmental Engineering, The University of Kansas, Lawrence, KS, USA.
✉e-mail: gvillari@princeton.edu

(structural) sets the general trends for hydrologic routing, but as hydrologic routing causes the transfer of matter from hills to valleys (functional) and reorganizes the landscape over centuries and millennia, new flow paths are created and a new structural arrangement emerges[17]. This idea of hillslope connectivity has made it a focal point of recent research on quantifying landscape dynamics due to its potential for improved management of water and environmental systems[1,18,19]. However, large-scale analysis of what controls structural connectivity has only recently been explored[13], and climate change-driven connectivity analyses have been limited to ecological connectivity[20,21].

For this study, we examine structural connectivity and climate due to the computational requirements to perform a large-scale functional connectivity analysis. We adopt a framework proposed by ref. 22 in which we focus on long-term catchment response to climate in regard to structural connectivity as a first step in modeling global hillslope connectivity. Before we can understand the role of climate change on structural hillslope connectivity, we need to first understand and model the drivers of these pathways. To explain, structural connectivity is informed by topographic features (e.g., elevation, slope, and roughness)[14] and these features record the history of the landscape as they are the result of tectonic and climatic processes[23]. Furthermore, processes such as tectonics[13] have been shown to play a crucial role in structural connectivity across large spatial domains. Despite these advances, we only have local (i.e., individual basins[15,16,18])

or regional[5,13,24] information about what drives connectivity, and we lack a global view of this phenomenon. Moreover, a model that allows capturing tectonic and climatic drivers worldwide is still missing, hindering our capability of making statements about future changes in hillslope connectivity.

In this work, we ask the questions: what are the drivers of global structural hillslope connectivity and how will climate change alter the connectedness of landscapes? To answer this question, we estimate structural hillslope connectivity for over 3500 basins across the globe at a spatial resolution of 90 m. We quantify the potential strength of structural hillslope connectedness with the Index of Connectivity (IC)[25] and develop a statistical model to explain its climatic and tectonic drivers (details in Methods). We evaluate our hypothesis on the effects of climate change on connectivity by applying the model to four future climate scenarios.

## Results and discussion

### Global assessment of connectivity at the hillslope scale
Basin-averaged structural hillslope connectivity estimates exhibit substantial spatial variability both within and across continental landmasses (Fig. 1). Connectivity is the highest near mountainous regions such as the Himalayas in Asia, the Alps in Europe, the Cascade and Rockies in North America, the Andes in South America, and the Ethiopian Highlands and Drakensburg in Africa. Additionally, higher connectivity is present in islands such as Japan, New Guinea, and New

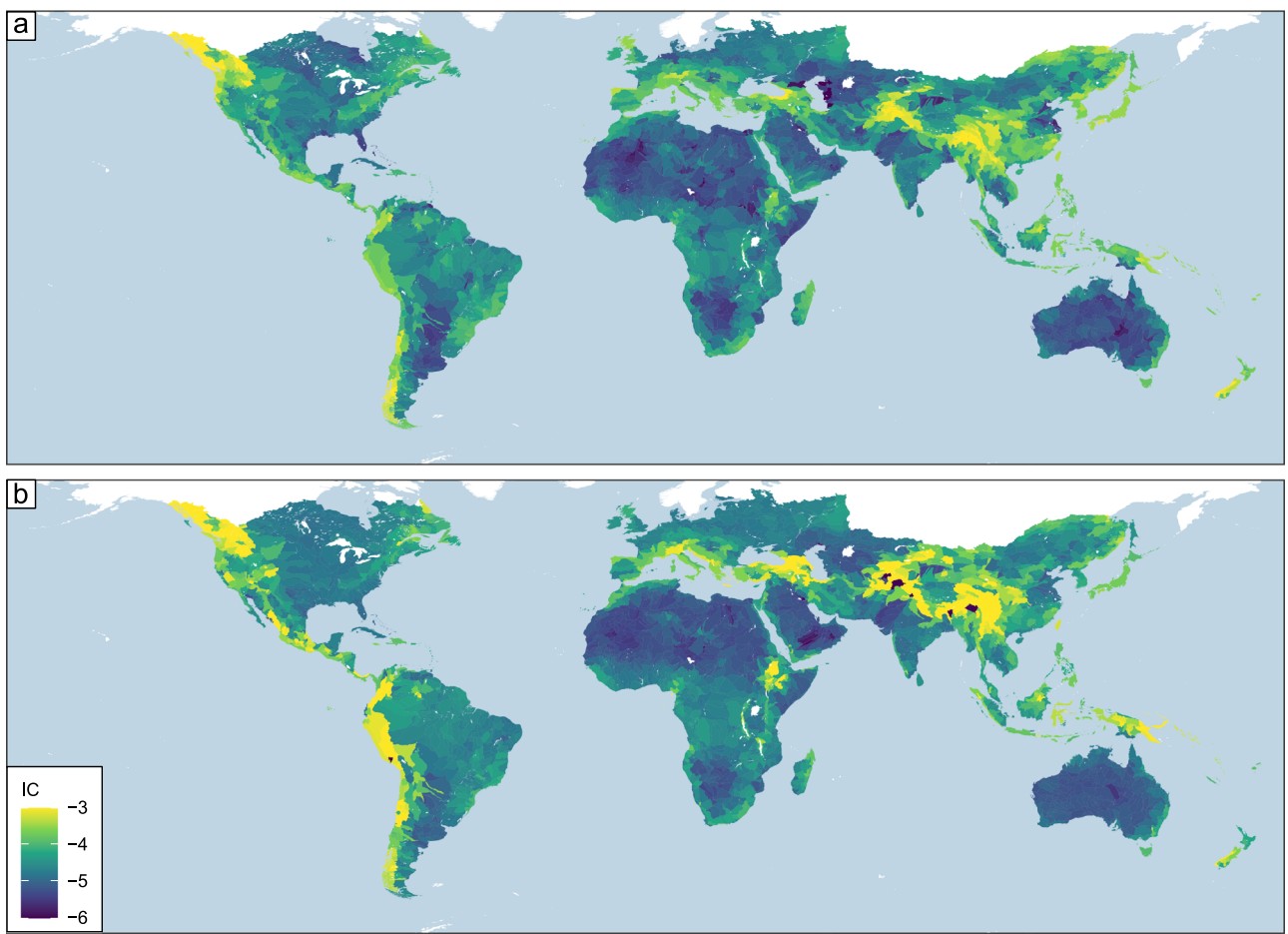

**Fig. 1 | Estimated and modeled basin-averaged index of connectivity (IC). a** The basin-averaged index of connectivity based on HydroSHEDS level 5 is determined from Eq. 1. **b** Basin-averaged IC modeled with a statistical model (Eqs. 6–9) utilizing precipitation, potential evapotranspiration, peak ground acceleration, mean river segment slope, river segment length, and mean elevation of the river profile segment as predictors. To provide context, IC can range from [−∞, ∞] and the smaller (more negative) a value, the less connected a basin is. For this plot, low connectivity is on the darker end of the spectrum (−6) and higher connectivity is at the lighter end of the spectrum (−3).

Zealand. The lowest connectivity values are observed in arid regions such as the Sahara and Kalahari deserts in Africa, Arabian and Syrian deserts in Western Asia, and the Great Australian desert in Australia. These areas highlight the nature of the DEM-based model, with large topographic relief driving connectivity. Based on the analyses by ref. 26, this spatial pattern shows a strong association with locations where converging tectonic plates are.

### Climatic and tectonic drivers of structural connectivity
To understand what is driving these patterns in connectivity, we now shift our focus to examining the correlation with various climatic and tectonic proxies. First, we examine the association with precipitation and potential evapotranspiration (PET) and find strong correlations (i.e., Spearman's rho values of 0.39 and −0.42, respectively). Figure 2 displays these correlations with these metrics in a scatter plot form, clearly showing the dependence between IC and the different potential drivers. These results suggest that the wetter basins (i.e., higher precipitation or lower PET) tend to have higher structural connectivity. Furthermore, we do not find a dependence between elevation and precipitation or PET, suggesting that the correlation between IC and climate variables is not mediated by elevation. A likely reason for the correlation we find between the climate variables and IC could be due to higher drainage densities in the wetter areas, providing a closer target for the IC calculation, leading to higher IC values on average. Additionally, these results differ from those by ref. 13 for the contiguous United States, which found a negative association for precipitation; the discrepancies are likely due to the differences in the range of precipitation (regional vs global), basin scale, river network

for targets (see ref. 27), and connectivity model resolution (10-m vs 90-m DEM). We find the relative importance of climatic variables increases compared to ref. 13 for the United States, but are still secondary to tectonics. Finally, it is important to note that we use climate variables of precipitation and PET over decadal time periods in our analyses. Structural connectivity is theorized to be set by landscape evolutions that require longer temporal scales[14], which highlights a temporal scale mismatch between drivers and supports our correlation results.

For tectonic drivers, we find stronger correlations with IC than with climate processes (Fig. 2). The highest correlation for tectonic proxies is the basin-averaged river profile slope[26,28] (Spearman $\rho = 0.84$). This confirms that when the primary river channel is steep, structural connectivity to the river network is high and matches the results from ref. 13. Peak ground acceleration is the next highest correlated value, with a positive correlation (Spearman $\rho = 0.47$). Basin-averaged river profile elevation[28] and profile length[28] have weaker correlations, with Spearman rho values of 0.33 and −0.40, respectively. Based on these univariate analyses, tectonics play a larger role than climate processes. Finally, the tectonic impact on structural connectivity is due to the creation of steep landscape settings: plate boundary convergences create mountainous areas where steep slopes provide the energy to transport material to nearby flatlands (Fig. 1).

### Climate change impacts to global structural connectivity
The analyses up to this point were based on computing the correlation between the connectivity index and each of the predictors. To assess the projected changes in IC due to climate change, we have developed a model in which IC is regressed against six predictors (i.e., elevation,

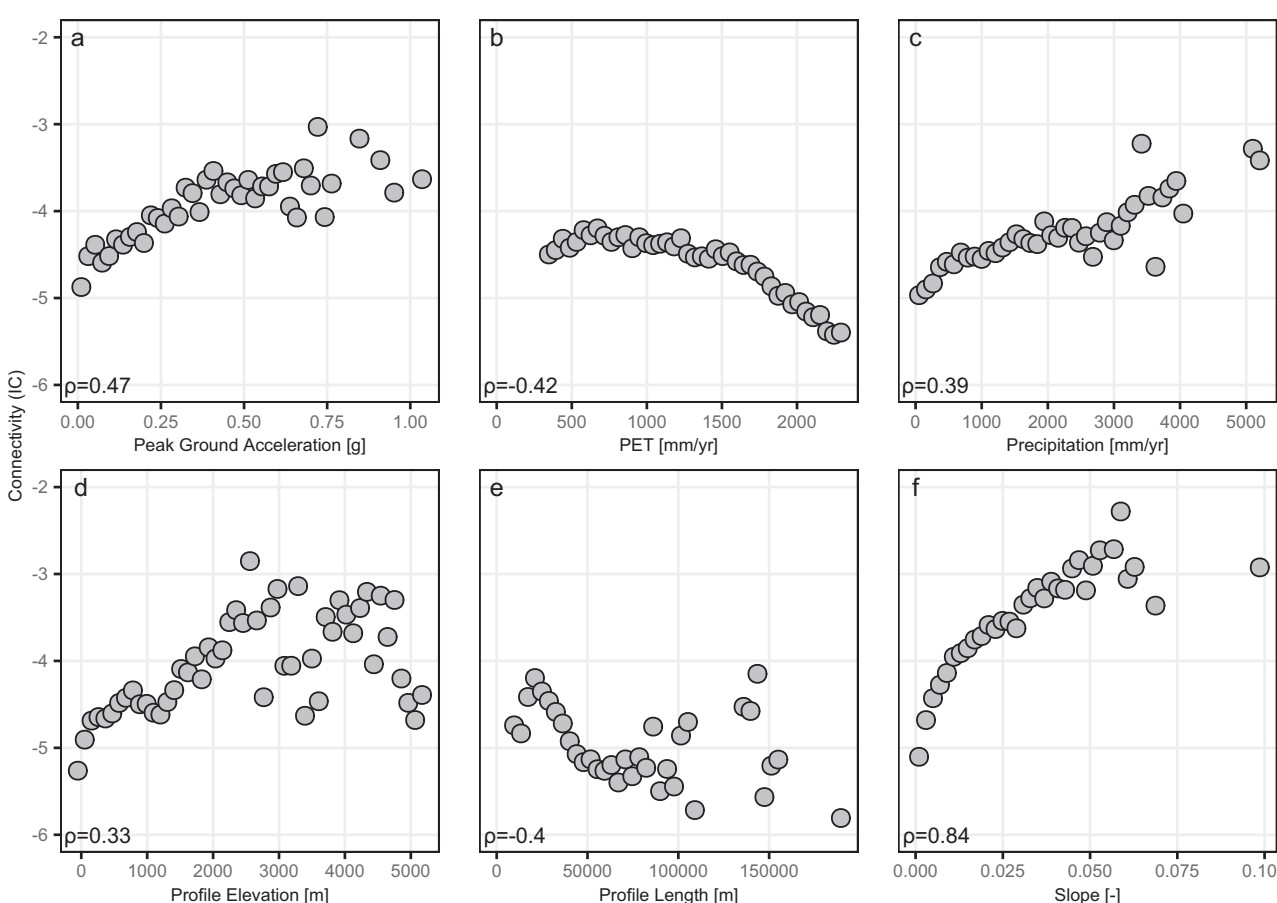

**Fig. 2 | Correlations between the index of connectivity and climatic and tectonic indices.** The indices are **a** peak ground acceleration, **b** potential evapotranspiration (PET), **c** precipitation, **d** profile elevation, **e** profile length, and **f** slope. The Spearman correlation is computed for all basin-averaged values with binned values (50 bins) shown on the plot.

PET, peak ground acceleration, precipitation, profile length, and slope) (see Methods for a description of the model and Fig. S1 for the correlation among the different predictors). We find slope is the only predictor that is selected for all four parameters of the Skew t type I distribution, with profile length and peak ground acceleration that are selected for three out of the four parameters (see equations within Methods). Elevation, precipitation, and PET are selected for two out of the four parameters. Consistent with the results in Fig. 2, the wetter the basin the larger the IC. Despite its simplicity, this model can reproduce the spatial variability in IC at the global scale, overall capturing the areas with higher/lower values (compare the top and bottom panels in Fig. 1). Based on the residuals' statistics, the model can explain the systematic signal in the data, with the residuals that are white noise, further supporting the goodness of fit of this model. Finally, in setting up our regression model for the projections, it is important to note that we are assuming the current landscape is in equilibrium with the current climate, and projected changes should be interpreted with this in mind.

In assessing the role of climate change, we utilize four shared-socioeconomic pathways (SSPs) provided by the Coupled Model Intercomparison Project Phase 6 (CMIP6) to represent different socioeconomic scenarios that couple greenhouse gas emissions with climate policies. Specifically, we focus on the impact of PET and precipitation on changes in connectivity. Overall, connectivity is projected to increase for ~56% of the basins across the globe, covering an area that is 55% (average across all SSPs) of the total land area considered here. Under SSP126 and SSP245 scenarios, the changes in IC are rather muted, with values of change of <2% globally. However, as the greenhouse gas emissions increase (i.e., SSP370 and SSP585), there are large areas of the globe where changes to structural connectivity become much more pronounced (Fig. 3). The most notable increase in structural connectivity under these two emission scenarios occur in southern Asia (Ganges Basin) and the Indonesia and Melanesia Islands, which are areas characterized by high historical average annual precipitation compared to the rest of the globe (Fig. S3). The greatest reductions in connectivity of up to 4% are expected to occur in the northeast (Venezuela, Guyana Suriname) and southern part (Chile) of South America as well as Central America and Central Africa. Finally, we find that large projected changes in climate variables align with large changes in IC for some regions of the globe. However, these results should be interpreted with caution as our statistical model does not explicitly capture region-specific lithology that is also crucial to landscape evolution.

## Projected connectivity impacts on global water, solute, and sediment fluxes

Water, solute, and sediment fluxes across the landscape are mediated by the availability of energy for transport, the activation of hydrologic pathways, and the impendence of buffers and other disconnectivities along the transport cascade. Recent work indicates that climate change will increase mass movement erosion across much of New Zealand, which will primarily be driven by storm magnitude frequency increases[29]. Our work indicates that this increase in erosion in New Zealand will be coupled with increases in hydrologic connectivity, thus potentially linking the production of sediment through erosion with the delivery to downstream waters through connectivity, resulting in potentially deleterious effects on riverine water quality. Across global cold regions, including the Arctic, Scandinavia, Patagonia, and the Cascades, atmospheric warming is expected to reduce the number of frozen areas of the Earth (cryosphere) and expose more sediment sources to potential erosive forces[30]. While our modeling approach does not capture many northern cold regions, the available results indicate that projected structural connectivity changes (i.e., increase or decrease) are not consistent with changes in sediment erosion. For example, in the Southern Andes of Patagonia, structural connectivity is

projected to decrease while ref. 30 indicates erosion is projected to increase. This means that, while more sediment is projected to be produced as the frozen areas thaw, the transport pathways to downstream waterbodies might decrease, resulting in intermediate deposition of eroded material. Analysis of observational data conducted by ref. 31 supports this concept as the authors found that glacier recession leads to increased connectivity between the upper basin and proglacial areas, but river reworking of glacial till and coarse sediment create negative feedbacks that reduce sediment export.

Overall, based on the correlation analyses we find that structural connectivity is controlled primarily by tectonic factors with climatic factors being second-order controls. Additionally, the coupling of a DEM-based connectivity metric like IC with a statistical model allowed us to understand the role of different drivers and how they are expected to shift the landscape in the future. Our results indicate structural connectivity, specifically the IC, is projected to change by as much as four percent depending on the emission scenario due to changes in precipitation and PET. These findings provide basic information in terms of hot spots that should be further considered to better quantify the impacts of climate change on these potential transport pathways and to improve watershed management. In regions of the world where structural connectivity is expected to increase or decrease substantially, these shifts in landscape interactions will cause changes in water, sediment, and nutrient transport that will impact agriculture, natural habitats, and flood control systems. However, the time scale at which climate change occurs might be too short to manifest itself in structural connectivity changes for the other regions of the world, but functional connectivity could be altered due to climate change as it is driven by climatic variables such as precipitation at short temporal scales[5,15,16]. Furthermore, in regions where we project similar changes in structural connectivity, functional connectivity could be altered in completely different ways and so our takeaways should not be extrapolated to short-term durations. Future works to model functional connectivity at similar scales should be conducted to expand upon our efforts and capture a complete picture of hillslope connectivity as structural connectivity alone is not enough (see ref. 32). Our study presents an initial step in identifying regions across the world where structural hillslope connectivity dominates and is projected to be impacted by climate change. We encourage stakeholders to utilize our findings to focus initial climate planning efforts related to large-scale watershed management practices.

## Methods

To represent structural hillslope connectivity across the globe, we calculated the IC based on the same methodology as ref. 13, which describes the probability of sediment from an upslope point traveling to a downslope target (streams). IC is empirically defined as:

$$IC = \log_{10}\left(\frac{D_{up}}{D_{dn}}\right) \quad (1)$$

where $D_{up}$ and $D_{dn}$ represent the upslope and downslope elements of connectivity. IC values can range from $[-\infty, \infty]$ with greater values indicating higher connectivity.

The upslope component, $D_{up}$, represents the potential of sediment yields from upslope sources to be routed downward and is defined as:

$$D_{up} = \bar{W}\bar{S}\sqrt{A} \quad (2)$$

where $\bar{W}$ is the average weighting factor of the contributing upslope area, $\bar{S}$ is the average slope of the contributing upslope area (m/m) and A is the contributing upslope area (m²).

Next, we define the downslope component, $D_{dn}$, as the probability of the sediment flow to travel along the flow path arriving at the

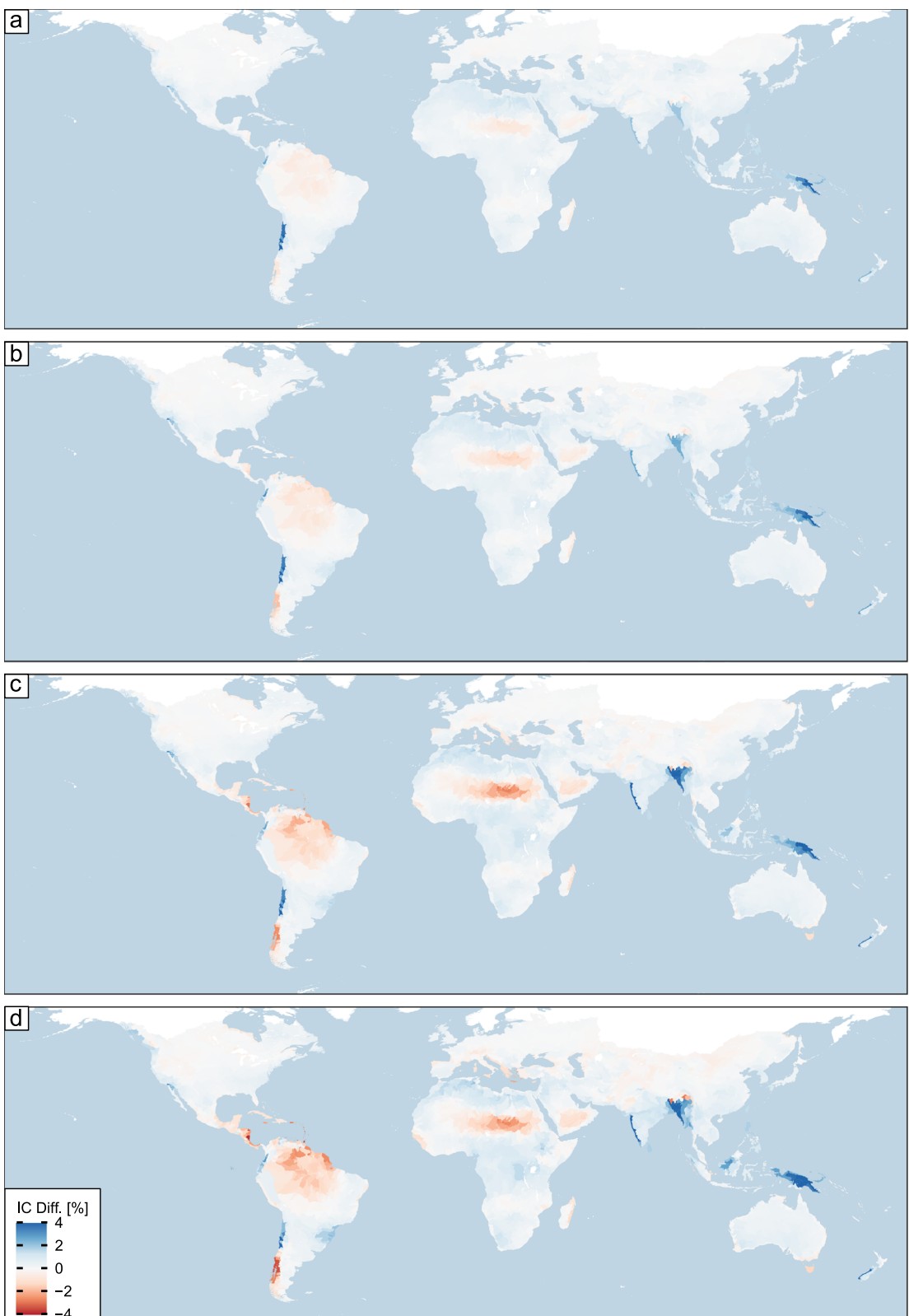

**Fig. 3 | Projected changes in the index of connectivity based on future scenarios.** Percent change in mean basin index of connectivity (IC) values based on changes in average annual precipitation and potential evapotranspiration between the future (2070–2100) and historical period (1970–2000) for Shared-Socioeconomic Pathways (SSPs) of **a** SSP126, **b** SSP2452, **c** SSP370, and **d** SSP585. Blue colors indicate an increase in connectivity, whereas red ones show a decrease (i.e., more disconnected).

nearest target. The downslope component is defined as:

$$D_{dn} = \sum_i \frac{d_i}{W_i S_i} \qquad (3)$$

where $d_i$ is the flow path length to the downstream channel for the $i$th cell at the steepest slope direction (m). $W_i$ and $S_i$ are the weighting factor and slope gradient, respectively, at the $i$th cell.

For our analysis, we determine the weighting factor, W, based on the roughness index (RI) or surface roughness as defined by ref. 18. RI is the standard deviation of the residual topography computed over a $5 \times 5$ cell moving window defined as:

$$RI = \sqrt{\frac{\sum_{i=1}^{25}(x_i - x_m)^2}{25}} \qquad (4)$$

where $x_m$ is the value of residual topography at the $i$th cell within the window and $x_m$ is the mean of the cells within the moving window. Finally, the weighting factor is defined as:

$$W = 1 - \left(\frac{RI}{RI_{max}}\right) \qquad (5)$$

where $RI_{max}$ is the maximum RI value for a region.

For inputs to calculate IC, we utilized DEMs with a 90-meter spatial resolution from HydroSHEDS[33]. To determine $d_i$ in Eq. 3, we defined targets as nearby streams. We utilized streams from the HydroRIVERS dataset[34]. Additionally, HydroSHEDS' 90-m DEM does not provide coverage above 60 °N so we do not conduct our analysis for the Arctic, Greenland, Iceland, Scandinavia, and Siberia. For the primary analysis, we utilized HydroBASINS[34] level 5 as the unit to conduct basin-averaged analyses. More specifically, for IC and the drivers we take the average value across the level 5 defined basin and run our analyses.

To examine the drivers of connectivity at a global level and its sensitivity to climate change, we built a regression model based on the Generalized Additive Model for Location, Scale, and Shape (GAMLSS)[35]. We utilized tectonic and climate drivers from refs. 26,28 as they computed the basin averages at the same HydroBASINS level. The tectonic proxies consist of the Peak Ground Acceleration (PGA) from the Global Earthquake Model GEM[36], mean river segment slope, river segment length, and mean elevation of the river profile segment. For climatic drivers, we utilized precipitation and evapotranspiration from the WorldClim dataset[37]. We examined the correlation between the drivers (predictors) using a correlogram (Fig. S1) and found weak dependence among these covariates. Other predictors of river concavity, total relief, and aridity index from the ref. 26,28 were excluded in our initial data analysis because of a strong correlation with other covariates (i.e., multicollinearity). Next, we applied stepwise model selection to choose the best model for our purposes using the Schwarz Bayesian Criterion (SBC)[38] as selection criterion. We examined multiple distributions (Table S1) and model configurations to select the best model based on SBC.

For this study, we utilize the 4-parameter Skew t type I (ST1) distribution with the parameters depending on the predictors. The probability distribution function (pdf) for the ST1 is given as:

$$f_Y(y|\mu,\sigma,\upsilon,\tau) = \begin{cases} \frac{c}{\sigma_0}\left[1+\frac{\upsilon^2 z^2}{\tau}\right]^{-(r+1)/2} & \text{if } y < \mu_0 \\ \frac{c}{\sigma_0}\left[1+\frac{z^2}{\upsilon^2\tau}\right]^{-(r+1)/2} & \text{if } y \geq \mu_0 \end{cases} \qquad (6)$$

where μ is the location shift parameter [−∞, ∞], σ is the scaling parameter [−∞, ∞], υ is the skewness parameter [−∞, ∞], τ is the kurtosis parameter [0, ∞], and $z = (y - \mu)/\sigma$. For our analysis, y is the basin-

averaged IC values across the globe [−∞, ∞]. More information on the distribution can be found at ref. 35. For each of the distribution parameters the following linear equations were derived:

$$\mu = -5.092 - \left(9.037 \times 10^{-5}\right)PET - (7.427 \times 10^{-1})PGA \\ + \left(2.981 \times 10^{-4}\right)P + (1.210 \times 10^2)S \qquad (7)$$

$$\ln(\sigma) = -2.476 + (1.078 \times 10^{-4})P + (2.314 \times 10^{-5})PL + (7.380 \times 10^1)S \qquad (8)$$

$$\upsilon = 1.486 - (2.410 \times 10^{-4})EL - (1.134 \times 10^{-3})PET \\ + (2.768)PGA - (1.921 \times 10^{-5})PL - (6.396 \times 10^1)S \qquad (9)$$

$$\ln(\tau) = -7.088 \times 10^{-1} + (7.614 \times 10^{-4})EL - (3.057)PGA \\ + (6.678 \times 10^{-5})PL + (2.139 \times 10^2)S \qquad (10)$$

where EL is elevation (m), PET is potential evapotranspiration (mm/yr.), PGA is peak ground acceleration ($g$), $P$ is precipitation (mm/yr.), PL is profile length (m), and S is slope (−) as described above. For the ST1, μ and υ have identity link functions whereas logarithmic link functions are used for σ and τ. The goodness of fit statistics for residuals consisting of mean, variance, coefficient of skewness, coefficient of kurtosis, and Filliben correlation have values of −0.043, 0.957, 0.029, 3.235, and 0.999, respectively. To calculate a mean IC value for a given basin, the basin average values for the inputs (e.g., PET, PGA) are plugged into Eqs. 7–10 and the estimated parameters are utilized to compute the 50th percentile from the ST1 (Eq. 6). These IC values are shown in Fig. 1.

To assess the impact of climate change we utilized precipitation and PET from 34 climate models as part of the Coupled Model Intercomparison Project Phase 6 (CMIP6)[39] and computed a ratio of change between historical (1970–2000) and future periods (2070–2100). We then multiply the historical (1970–2000) annual average precipitation (Fig. S2) and PET (Fig. S3) for each basin by the ratio and recalculate IC based on our regression model (Eqs. 6–10). We provide the ratios of change in Fig. S4 (precipitation) and Fig. S5 (PET). We chose these two predictors for our sensitivity analysis as the time scale of which changes in climate occur is much smaller compared to tectonic drivers.

The primary limitations of our study are related to the features in the model. The initial calculation of IC based on DEMs is dependent upon the provided stream targets and artifacts captured in the DEM. First, our IC estimates may be overestimated as the HydroRIVERS dataset uses a constant flow accumulation threshold of 100 cells where channels could be shown to exist that are not actually there. Future work should examine the influence of channel density on connectivity similar to ref. 27 with newer river datasets (e.g., ref. 40). We utilized a roughness factor described by ref. 18 for the weighting factor described above but this may not be adequate to capture topographic roughness when utilizing a 90-m DEM. For future studies, we suggest the exploration of utilizing the Universal Soil Loss Equation C-factor in place of RI as shown in refs. 25,41. This means that we did not capture the role of land use changes on structural connectivity in the future where transitions to urban or agricultural lands could offset changes due to climate. Further analyses of the impact of climate change should be performed regarding functional or dynamic connectivity, which utilizes physically based models to understand hillslope response[15,16]. Functional connectivity is driven by shorter precipitation events that are expected to change in magnitude and frequency in the future[42]. The use of physically based models could also aid with making causal statements beyond the correlation results presented here.

Finally, due to the larger time scales of tectonic actions, we did not explore the sensitivity of these parameters. However, future modeling efforts should explore these drivers to understand exactly the time scale at which structural connectivity is impacted.

## Data availability

All data used in this study is publicly available. Climate model data were downloaded from the WCRP Coupled Model Intercomparison Project (Phase 6) data portal found at https://esgf-node.llnl.gov/search/cmip6/. HydroSHEDS data can be found at https://www.hydrosheds.org/products. The IC results produced in this study for 90-m resolution at the basin scale are provided in the repository found in the code availability statement.

## Code availability

All code for the analysis and figures is available at https://doi.org/10.4211/hs.cc08f5fb62b54d29943dcc1da5df6b42.

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

## Acknowledgements
We acknowledge Stefano Crema and Marco Cavalli for making their IC tool, SedInConnect, and its code open source. This publication was supported by the Princeton University Library Open Access Fund.

## Author contributions
A.M. contributed to the conceptualization, formal analysis, and writing of the manuscript. G.V. contributed to the conceptualization, methodology, and writing of the manuscript as well as supervised the work. A.H. contributed to the review and editing of the manuscript.

## Competing interests
The authors declare no competing interests.
