## [Peer Review File · Nature Communications]

Climate change projected to impact structural hillslope connectivity at the global scaleREVIEWER COMMENTS

Reviewer #1 (Remarks to the Author):

dear Authors.

The issue of evaluating the connectivity of flows on a continental and global scale is a topic of absolutely great interest in the scientific community in the geomorphological and hydrological area. Therefore, the article should be fully considered, due to its theme and scale of application, in line with the editorial objectives of the magazine.

The presented manuscript has a fairly good writing style, however in the points indicated below it could be improved to make reading easier and more understandable for readers.

Here are some suggestions to improve the manuscript in its style, readability and content.

1) in the abstract section always clearly indicate whether in the geographical macro-areas indicated as an example an increase or a decrease in connectivity is expected following the expected climate change. Also indicate the time period, in the future, considered in the scenario analysis.

2) in the "main text" section, give a concise definition of structural connectivity and describe the difference with functional or dynamic connectivity which is mentioned after line 204.

3) in the "projected connectivity..." section, rewrite the text between lines 144 and 151 in a simpler form. Better to avoid the term "de-coupling". a simpler text would be better. The concepts expressed are extremely important and fundamental for future studies on the subject.

4) "methods" section. explain in more detail the methodology for calculating the IC index considering the starting data used. In particular the resolution of the global DEM. Better explain the calculation method of IC referred to as "stream Target". Better explain what is referred to as "basin average analysis"

5) in table 1 it is convenient to rewrite and rearrange the data and the coefficients since it is not clear how the coefficients are used in the derived models. It is not clear whether these are linear or non-linear correlation models. The symbols used in the table are not described. at least one example of the model obtained should be presented in an extended way in order to make the structure of the derived models clearer. some additional note in the text, as comment to the table, must be provided.

Reviewer #2 (Remarks to the Author):

Sediment connectivity is an important system property governing the efficiency of sediment transfer through catchments.

Hence, it also mediates the propagation of local-scale changes (e.g. hillslope-scale erosion) to larger spatial scales; for example, enhanced erosion could lead to sediment-related problems in downstream sections in areas of high connectivity, while the eroded sediments would tend to be deposited near their source where connectivity is poor. Connectivity has two aspects, namely structural and functional connectivity. Both components are subject to changes, and it is the aim of the present study to predict possible impacts of climate change on "hillslope connectivity".

The present study is the first to compute the well-established index of connectivity IC at a near-global scale; the authors were also the first to employ that index on a continental (CONUS) scale in their 2022 GRL publication. In order to investigate possible changes to (structural) connectivity, the authors first correlate IC at the catchment scale with a number of climatic and tectonic proxies. Then, regression models of IC on six topographic climatic variables are set up and then applied in order to predict how connectivity (better: the IC) could change given four climate change scenarios.

The reviewer fully agrees with the stated importance of connectivity in propagating climate change impacts, and also with the statement that connectivity itself is subject to change. Therefore, I think that the topic is of high interest for the geomorphological community; moreover, connectivity is important for catchment management, including natural hazards.

I am impressed by the spatial extent (and the computational challenges associated with it) of this work. The large spatial scale makes it necessary to accept deficiencies, for example it is questionable whether it is viable to compute the impedance component W from "microtopographic undulations" if the DEM used has a spatial resolution of 90m; my personal take on this would have been to use the RUSLE C factor based on global landcover data. Another compromise that could be acceptable is the use of a purely topography-based global channel network dataset (although there seem to be recent datasets that take into account different channel densities). In all I think that, technically, the computation of indices, correlations and the sophisticated regression models is appropriate.

The correlation analysis yields interesting results, for example the strong influence of tectonic proxies on structural connectivity. Looking at the correlation between IC and climatic variables, however, I have strong concerns that correlation and causality might have been (partially) confused, especially looking at the next step (the regression model used for predictions): It is absolutely plausible that IC is high in mountain areas because of their steepness (and that makes a causal relationship with tectonic proxies plausible), and because of the density of valleys/channels. But the statement "connectivity is higher in wetter regions" is not necessarily a causative one (precipitation causes higher structural connectivity) but could be a simple coincidence: Mountains cause higher precipitation, and at the same time IC is higher for the reasons explained above. This note of caution has probably not been considered by the authors, and I feel it should be.

Another major concern is the way climate change could influence structural connectivity; I think it is very plausible to assume that enhanced magnitude and/or frequency of heavy rain, for example, has a strong and more or less immediate impact on runoff formation and functional connectivity. But the reaction of topography and vegetation (factors of structural connectivity) takes by far longer time, and is less "direct" than the reaction of functional connectivity. The focus of the present study clearly is structural connectivity, and I think that the very different temporal scales should be discussed far more carefully: The observed correlation of IC and climatic variables, if not coincidence (as mentioned above), would be the results of tens to hundreds of thousands of years in order to reach an equilibrium of climatic forcing and topography. Looking at comparatively swift recent climatic changes, the response of structural connectivity to these changes can be expected to be far slower than the more or less direct response of functional connectivity. The latter, however, is by no means included in the structural IC index, and I don't know a single study that validates IC beyond field observation on the single catchment scale. Moreover, two identical catchments (topography-wise, and hence also with respect to their structural connectivity/IC) can regularly experience vastly different functional connectivity depending on hydrometeorological forcing. While the authors do acknowledge the importance of modelling to investigate the reaction of functional connectivity to changes in forcing, the lack of discussion regarding the reasons for the observed correlation and also regarding the temporal scale of structural connectivity changes has led to predictions that could be valid only on long to very long time scales and can be assumed to be less important than the changes in functional connectivity, that cannot be predicted using static SC indices like IC alone.

I have added more thoughts, comments and suggestions in the annotated PDF attached to this review; I hope that the comments help to understand my thoughts and concerns.

Reviewer #3 (Remarks to the Author):

Review of "Climate change amplifies structural hillslope connectivity at the global scale" by Michalek et al.

This manuscript presents a new global (excluding high latitudes) dataset of connectivity indices and then explores the correlation of that connectivity index with various topographic, climatic, and seismic datasets. The manuscript then introduces a statistical model to predict connectivity index and attempts to forecast the impact of climate predictions (precipitation and evapotranspiration) on connectivity. The study finds that the connectivity index is more strongly correlated with topographic and seismic properties than with climate properties, but still shows some correlation with precipitation and evapotranspiration (significance was not evaluated). By applying forecasted precipitation and evapotranspiration, the study predicts that about half of all basin will have greater connectivity (and presumably half of all basins will have lower connectivity).

The topic is an interesting one to explore, but paper unfortunately does a very poor job of explaining what was actually done, making it impossible to evaluate the work or understand its implications. At the most fundamental level, it is not clear from this manuscript what is meant by "connectivity" or how it was measured. The first paragraph motivates the study by highlighting the importance of connectivity within the stream and river network, and the next paragraph moves into connectivity between hillslopes and the stream network. These paragraphs also discuss connectivity of water, sediment, and nutrients, which are all related but driven by different sets of processes. It's not clear to me which form of connectivity is being evaluated here, and there is no explanation of how it is measured. How can the reader interpret a connectivity index of -4 without any explanation?

Moreover, the finding that is claimed in the title is not properly evaluated in the manuscript. Based on my reading of the work, about half of all basins are predicted to have greater connectivity and about half of all basins are expected to have lower connectivity under various climate scenarios. In other words, the impact of climate change on connectivity is a wash. I may be reading that incorrectly, but if so, the paper needs to make that clear, starting with a proper evaluation of the significance of predicted impact of climate change on connectivity.

Also, without knowing how "structural connectivity" is calculated, I'm left wondering how flexible it is to changes in climate. The "structural" part of it makes me think that it's a topographic measure. If that's true, the impact of climate on structural connectivity would require topography to adjust to the new climatic regime before those connectivity adjustments are realized.

I believe that there could be some interesting findings here but the presentation needs a substantial amount of work so that the reader can understand what's going on.

Comments by line number

70. I assume "changes in elevation" refers to "topographic relief" and not temporal evolution

83. "importance of climatic variables increases" with respect to what?

85. "longer-term climate variables" What are these? Were they evaluated?

89. "drivers" should be "proxies"

89. "river profile slope" over which spatial scale and on which segments?

92. "river profile elevation" Elevation of which part of the river?

113 - . Good to state in the main text which model parameters you're tweaking from the climate model and not require the reader to go to the methods.

127 - 133. I don't understand what is trying to be said here. It's not clear from the manuscript how to intuitively relate changes in climate to connectivity.

Reviewer #1 (Remarks to the Author):

dear Authors.

The issue of evaluating the connectivity of flows on a continental and global scale is a topic of absolutely great interest in the scientific community in the geomorphological and hydrological area. Therefore, the article should be fully considered, due to its theme and scale of application, in line with the editorial objectives of the magazine.

The presented manuscript has a fairly good writing style, however in the points indicated below it could be improved to make reading easier and more understandable for readers.

Here are some suggestions to improve the manuscript in its style, readability and content.

1) in the abstract section always clearly indicate whether in the geographical macro-areas indicated as an example an increase or a decrease in connectivity is expected following the expected climate change. Also indicate the time period, in the future, considered in the scenario analysis.

R: Thank you for the suggestion. We have clearly indicated whether a projected increase or decrease in connectivity is expected for the regions listed within the revised manuscript.

Please see lines 22 to 26 in the revised text as follows:

“We find climate change is projected to alter global-scale connectivity at the end of the century (2070 to 2100) by up to 4% for increasing greenhouse gas emission scenarios. Notably, the Ganges River, the world’s most populated river basin, is projected to experience a drastic increase in connectivity. Conversely, the Amazon River and the Pacific coast of Patagonia are projected to experience the largest decreases in connectivity.”

2) in the "main text" section, give a concise definition of structural connectivity and describe the difference with functional or dynamic connectivity which is mentioned after line 204.

R: Thank you for the comment. We have revised the first section of the main text when defining hillslope connectivity, by adding specific definitions of structural and functional connectivity.

The changes have been made as follows in lines 54 to 87:

“ Hillslope connectivity refers to the linkage of upstream sources and downstream transport pathways (e.g., rills, gullies, rivers) and is informed by topographic features, which themselves encode the tectonic and climatic history of a landscape^{1,2}. Specifically, hillslope connectivity is composed of two interconnected types of connectivity, structural and functional³. Structural (“static”) connectivity describes the spatial arrangement of the system components, which established the long-term potential for downstream transport^{4,5}. Functional (“dynamic”) connectivity is the interplay of spatial and temporal fluxes within the system for the short-term (i.e., storm event response)^{6,7}. The coevolution can be expressed as the initial landscape arrangement (structural) sets the general trends for hydrologic routing, but as hydrologic routing causes the transfer of matter from hills to valleys (functional) and reorganizes the landscape over

centuries and millennia, new flow paths are created and a new structural arrangement emerges⁸. This idea of hillslope connectivity has made it a focal point of recent research on quantifying landscape dynamics due to its potential for improved management of water and environmental systems⁹⁻¹¹. However, large-scale analysis of what controls structural connectivity has only recently been explored⁴ and climate change driven connectivity analyses have been limited to ecological connectivity^{12,13}.

For this study, we examine structural connectivity and climate due to the computational requirements to perform a large-scale functional connectivity analysis. We adopt a framework proposed by Ref¹⁴ in which we focus on long-term catchment response to climate in regard to structural connectivity as a first step in modeling of global hillslope connectivity. Before we can understand the role of climate change on structural hillslope connectivity, we need to first understand and model the drivers of these pathways. To explain, structural connectivity is informed by topographic features (e.g., elevation, slope, and roughness)⁵ and these features record the history of the landscape as they are the result of tectonic and climatic processes¹⁵. Furthermore, processes such as tectonics⁴ have been shown to play a crucial role on structural connectivity across large spatial domains. Despite these advances, we only have local (i.e., individual basins^{6,7,9}) or regional^{4,16,17} information about what drives connectivity, and we are lacking a global view of this phenomenon. Moreover, a model that allows capturing tectonic and climatic drivers worldwide is still missing, hindering our capability of making statements about future changes in hillslope connectivity.”

3) in the "projected connectivity...." section, rewrite the text between lines 144 and 151 in a simpler form. Better to avoid the term "de-coupling". a simpler text would be better. The concepts expressed are extremely important and fundamental for future studies on the subject.

R: Thank you for the comment and we agree it should be made easier to understand. We have changed this section as suggested.

The changes can be found as follows in lines 205 to 242. Note we also made changes to address comments from Reviewer 2 as well that are shown.

“..... Across global cold regions, including the Arctic, Scandinavia, Patagonia, and the Cascades, atmospheric warming is expected to reduce the amount of frozen areas of the Earth (cryosphere), and expose more sediment sources to potential erosive forces¹⁸. While our modeling approach does not capture many northern cold regions, the available results indicate that projected structural connectivity changes (i.e., increase or decrease) are not consistent with changes in sediment erosion. For example, in the Southern Andes of Patagonia, structural connectivity is projected to decrease while Ref¹⁸ indicates erosion is projected to increase. This means that while more sediment is projected to be produced as the frozen areas thaw, the transport pathways to downstream waterbodies might decrease, resulting in intermediate deposition of eroded material. Analysis of observational data conducted by Ref¹⁹ supports this concept as the authors found that glacier recession leads to increased connectivity between the upper basin and proglacial areas but river reworking of glacial till and coarse sediment create negative feedbacks that reduce sediment export.”

4) "methods" section. explain in more detail the methodology for calculating the IC index considering the starting data used. In particular the resolution of the global DEM. Better explain the calculation method of IC referred to as "stream Target". Better explain what is referred to as "basin average analysis"

R: Thank you for the comment. We have expanded the methods section in the revised manuscript. First, we have added the IC calculation equations to clarify what was implemented. Next, we discussed the inputs and expanded their description. For the global DEM we utilized the HydroSHEDs product with a spatial resolution of 90-meter grid cell size. With the added equations we have defined that the stream target is the point in which connectivity is computed. For this we utilize the nearest stream segment a cell drains to as the target in which to compute the connectivity index. Finally, for basin average analyses, we computed the average IC value at the Hydrosheds level 5 and then conducted our analyses based on these average values to better visualize our findings.

We have added these points in the method section in lines 273 to 305 as follows:

“To represent structural hillslope connectivity across the globe, we calculated the Index of Connectivity (IC) based on the same methodology as Ref⁴, which describes the probability of sediment from an upslope point to travel to a downslope target (streams). IC is empirically defined as:

$$IC = \log_{10}\left(\frac{D_{up}}{D_{dn}}\right) \quad (1)$$

where D_{up} and D_{dn} represent the upslope and downslope elements of connectivity. IC values can range from $[-\infty, \infty]$ with greater values indicating higher connectivity.

The upslope component, D_{up} , represents the potential of sediment yields from upslope sources to be routed downward and is defined as:

$$D_{up} = \bar{W}\bar{S}\sqrt{A} \quad (2)$$

where \bar{W} is the average weighting factor of the contributing upslope area, \bar{S} is the average slope of the contributing upslope area (m/m) and A is the contributing upslope area (m²).

Next, we define the downslope component, D_{dn} , as the probability of the sediment flow to travel along the flow path arriving to the nearest target. The downslope component is defined as:

$$D_{dn} = \sum_i \frac{d_i}{W_i S_i} \quad (3)$$

where d_i is the flow path length to the downstream channel for the i^{th} cell at the steepest slope direction (m). W_i and S_i are the weighting factor and slope gradient, respectively, at the i^{th} cell.

For our analysis we determine the weighting factor, W , based on the roughness index (RI) or surface roughness as defined by Ref⁹. RI is the standard deviation of the residual topography computed over a 5×5 cell moving window defined as:

$$RI = \sqrt{\frac{\sum_{i=1}^{25} (x_i - x_m)^2}{25}} \quad (4)$$

where x_i is the value of residual topography at the i^{th} cell within the window and x_m is the mean of the cells within the moving window. Finally, the weighting factor is defined as:

$$W = 1 - \left(\frac{RI}{RI_{max}} \right) \quad (5)$$

where RI_{max} is the maximum RI value for a region.

For inputs to calculate IC, we utilized DEMs with a 90-meter spatial resolution from HydroSHEDS²⁰. To determine d_i in Eq. 3, we defined targets as nearby streams. We utilized streams from the HydroRIVERS dataset²¹. Additionally, HydroSHEDS' 90-m DEM does not provide coverage above 60° N and so we do not conduct our analysis for Arctic, Greenland, Iceland, Scandinavia, and Siberia. For the primary analysis we utilized HydroBASINS²¹ level 5 as the unit to conduct basin-averaged analyses. More specifically, for IC and the drivers we take the average value across the level 5 defined basin and ran our analyses.”

5) in table 1 it is convenient to rewrite and rearrange the data and the coefficients since it is not clear how the coefficients are used in the derived models. It is not clear whether these are linear or non-linear correlation models. The symbols used in the table are not described. at least one example of the model obtained should be presented in an extended way in order to make the structure of the derived models clearer. some additional note in the text, as comment to the table, must be provided.

R: Thank you for the comment. We have added the equations for the probability density function as well as the parameters to the Methods section, with an expanded description of how they are utilized.

We have added these points in the method section in lines 306 to 355 as follows:

“To examine the drivers of connectivity at a global level and examine its sensitivity to climate change, we built a regression model based on the Generalized Additive Model for Location, Scale and Shape (GAMLSS)²². We utilized tectonic and climate drivers from Refs^{23,24} as they computed the basin averages at the same HydroBASINS level. The tectonic proxies consist of the Peak Ground Acceleration (PGA) from the Global Earthquake Model GEM²⁵, mean river segment slope, river segment length, and mean elevation of river profile segment. For climatic drivers, we utilized precipitation and evapotranspiration from the WorldClim dataset²⁶. We examined the correlation between the drivers (predictors) using a correlogram (Fig. S1) and found weak dependence among these covariates. Other predictors of river concavity, total relief, and aridity index from the Ref^{23,24} were excluded in our initial data analysis because of strong correlation with other covariates (i.e., multicollinearity). Next, we applied stepwise model selection to choose the best model for our purposes using the Schwarz Bayesian Criterion (SBC)²⁷ as selection criterion. We examined multiple distributions (Table S1) and model configurations to select the best model based on SBC.

For this study we utilize the 4-parameter Skew t type I (ST1) distribution with the parameters depending on the predictors. The probability distribution function (pdf) for the ST1 is given as:

$$f_Y(y | \mu, \sigma, v, \tau) = \begin{cases} \frac{c}{\sigma_0} \left[1 + \frac{v^2 z^2}{\tau} \right]^{-(r+1)/2} & \text{if } y < \mu_0 \\ \frac{c}{\sigma_0} \left[1 + \frac{z^2}{v^2 \tau} \right]^{-(r+1)/2} & \text{if } y \geq \mu_0 \end{cases} \quad (6)$$

where μ is the location shift parameter $[-\infty, \infty]$, σ is the scaling parameter $[-\infty, \infty]$, v is the skewness parameter $[-\infty, \infty]$, τ is the kurtosis parameter $[0, \infty]$, and $z = (y - \mu)/\sigma$. For our analysis y is the basin averaged IC values across the globe $[-\infty, \infty]$. More information on the distribution can be found at Ref²². For each the distribution parameters the following linear equations where derived:

$$\mu = -5.092 - (9.037 \times 10^{-5})PET - (7.427 \times 10^{-1})PGA + (2.981 \times 10^{-4})P + (1.210 \times 10^2)S \quad (7)$$

$$\ln(\sigma) = -2.476 + (1.078 \times 10^{-4})P + (2.314 \times 10^{-5})PL + (7.380 \times 10^1)S \quad (8)$$

$$v = 1.486 - (2.410 \times 10^{-4})EL - (1.134 \times 10^{-3})PET + (2.768)PGA - (1.921 \times 10^{-5})PL - (6.396 \times 10^1)S \quad (9)$$

$$\ln(\tau) = -7.088 \times 10^{-1} + (7.614 \times 10^{-4})EL - (3.057)PGA + (6.678 \times 10^{-5})PL + (2.139 \times 10^2)S \quad (10)$$

where EL is elevation (m), PET is potential evapotranspiration (mm/yr.), PGA is peak ground acceleration (g), P is precipitation (mm/yr.), PL is profile length (m), and S is slope (-) as described above. For the ST1, μ and v have identity link functions whereas logarithmic link functions are used for σ and τ . The goodness of fit statistics for residuals consisting of mean, variance, coefficient of skewness, coefficient of kurtosis, and Filliben correlation have values of -0.043, 0.957, 0.029, 3.235, and 0.999, respectively. To calculate a mean IC value for a given basin, the basin average values for the inputs (e.g., PET , PGA) are plugged into eqs. 7-10 and the estimated parameters are utilized to compute the 50th percentile from the ST1 (eq. 6). These IC values are shown in Fig. 1.”

Reviewer #2 (Remarks to the Author):

Sediment connectivity is an important system property governing the efficiency of sediment transfer through catchments.

Hence, it also mediates the propagation of local-scale changes (e.g. hillslope-scale erosion) to larger spatial scales; for example, enhanced erosion could lead to sediment-related problems in downstream sections in areas of high connectivity, while the eroded sediments would tend to be deposited near their source where connectivity is poor. Connectivity has two aspects, namely structural and functional connectivity. Both components are subject to changes, and it is the aim of the present study to predict possible impacts of climate change on "hillslope connectivity".

The present study is the first to compute the well-established index of connectivity IC at a near-global scale; the authors were also the first to employ that index on a continental (CONUS) scale in their 2022 GRL publication. In order to investigate possible changes to (structural) connectivity, the authors first correlate IC at the catchment scale with a number of climatic and tectonic proxies. Then, regression models of IC on six topographic climatic variables are set up and then applied in order to predict how connectivity (better: the IC) could change given four climate change scenarios.

The reviewer fully agrees with the stated importance of connectivity in propagating climate change impacts, and also with the statement that connectivity itself is subject to change. Therefore, I think that the topic is of high interest for the geomorphological community; moreover, connectivity is important for catchment management, including natural hazards.

R: We thank the Reviewer for the in-depth feedback provided and have listed all of the changes made to address your comments below.

I am impressed by the spatial extent (and the computational challenges associated with it) of this work. The large spatial scale makes it necessary to accept deficiencies, for example it is questionable whether it is viable to compute the impedance component W from "microtopographic undulations" if the DEM used has a spatial resolution of 90m; my personal take on this would have been to use the RUSLE C factor based on global landcover data. Another compromise that could be acceptable is the use of a purely topography-based global channel network dataset (although there seem to be recent datasets that take into account different channel densities). In all I think that, technically, the computation of indices, correlations and the sophisticated regression models is appropriate.

R: Thank you for the comment. We have addressed the RUSLE C factor comment by adding it as a limitation and suggested its use future work.

The following has been added at lines 369 to 392:

“..... We utilized a roughness factor described by Ref⁹ for the weighting factor described above but this may not be adequate to capture topographic roughness when utilizing a 90-m DEM. For future studies, we suggest the exploration of utilizing the Universal Soil Loss Equation C-factor in place of RI as shown in Refs^{28,29}. This means that we did not capture the role of land use changes

on structural connectivity in the future where transitions to urban or agricultural lands could offset changes due to climate.”

The correlation analysis yields interesting results, for example the strong influence of tectonic proxies on structural connectivity. Looking at the correlation between IC and climatic variables, however, I have strong concerns that correlation and causality might have been (partially) confused, especially looking at the next step (the regression model used for predictions): It is absolutely plausible that IC is high in mountain areas because of their steepness (and that makes a causal relationship with tectonic proxies plausible), and because of the density of valleys/channels. But the statement "connectivity is higher in wetter regions" is not necessarily a causative one (precipitation causes higher structural connectivity) but could be a simple coincidence: Mountains cause higher precipitation, and at the same time IC is higher for the reasons explained above. This note of caution has probably not been considered by the authors, and I feel it should be.

R: Thank you for the note. We did explore the relationship between the elevation of the catchments and the precipitation as well as PET and did not find a strong signal across the globe. In the mountainous regions we examined, there tends to be higher elevations across the basin, but as shown in the plot below, there is no clear pattern with precipitation (or PET). However, as you see when you make the same plot for IC vs precipitation there is a much clearer (positive) relationship that was shown within the Spearman's rho. We do agree with the specific comment on the HydroSHEDS river network and have added a statement of caution within this instance as well as to the limitations paragraph. As far as correlation versus causation, we agree with the Reviewer and added text to make this issue clearer in the revised manuscript.

The following has been added at lines 131 to 133:

“.... Furthermore, we do not find a dependence between elevation and precipitation and PET, suggesting that the correlation between IC and climate variables are not mediated by elevation.”

Figure R1. The plot represents a 2D binned heat map of all Hydroshed level 5 basin average values for 90-m elevation (y-axis) and WorldClim potential evapotranspiration and precipitation (x-axis).

Figure R2. The plot represents a 2D binned heat map of all Hydroshed level 5 basin average IC values (y-axis) versus basin precipitation (x-axis).

Another major concern is the way climate change could influence structural connectivity; I think it is very plausible to assume that enhanced magnitude and/or frequency of heavy rain, for example, has a strong and more or less immediate impact on runoff formation and functional connectivity. But the reaction of topography and vegetation (factors of structural connectivity) takes by far longer time, and is less "direct" than the reaction of functional connectivity. The focus of the present study clearly is structural connectivity, and I think that the very different temporal scales should be discussed far more carefully: The observed correlation of IC and climatic variables, if not coincidence (as mentioned above), would be the results of tens to hundreds of thousands of years in order to reach an equilibrium of climatic forcing and topography. Looking at comparatively swift recent climatic changes,

the response of structural connectivity to these changes can be expected to be far slower than the more or less direct response of functional connectivity. The latter, however, is by no means included in the structural IC index, and I don't know a single study that validates IC beyond field observation on the single catchment scale. Moreover, two identical catchments (topography-wise, and hence also with respect to their structural connectivity/IC) can regularly experience vastly different functional connectivity depending on hydrometeorological forcing. While the authors do acknowledge the importance of modelling to investigate the reaction of functional connectivity to changes in forcing, the lack of discussion regarding the reasons for the observed correlation and also regarding the temporal scale of structural connectivity changes has led to predictions that could be valid only on long to very long time scales and can be assumed to be less important than the changes in functional connectivity, that cannot be predicted using static SC indices like IC alone.

R: We agree with the Reviewer that functional and structural connectivity changes need to be carefully distinguished from one another. The Reviewer is correct that climate change will have large impacts to functional/dynamic connectivity, for example with more extreme rainfall events and changes and the intensification of droughts. However, climate impacts to functional connectivity will ultimately feed forward into alterations of structural connectivity (and vice versa). In a recent paper, Ref⁸ couple a hydrologic and landscape model to show the coevolution of the structural and functional properties of a basin, as they relate to runoff generation and water storage. The initial structural template sets the general trends for hydrologic routing, but as hydrologic routing causes the transfer of matter from hills to valleys and reorganizes the landscape over centuries and millennia, new flow paths are created and a new structural template emerges. We propose a similar approach for the interactions of structural and functional connectivity in our analysis.

As the Reviewer states, functional changes can be rapid while structural changes are slower. In fact, our study shows precisely this: slight structural connectivity changes in most areas of the world (on average <2%) over the next 70 years. This is in contrast to the large functional meteorological change that will occur (e.g., precipitation in central Africa may increase by as much as 50%). Thus, while we do predict structural changes, their net change is slight relative to functional changes, which we would expect given the more gradual change of topographic element compared to hydroclimatic ones. We discuss the coevolution of structural and functional connectivity in greater detail in the revised manuscript.

The following has been added at lines 54 to 87 in the introduction:

“ Hillslope connectivity refers to the linkage of upstream sources and downstream transport pathways (e.g., rills, gullies, rivers) and is informed by topographic features, which themselves encode the tectonic and climatic history of a landscape^{1,2}. Specifically, hillslope connectivity is composed of two interconnected types of connectivity, structural and functional³. Structural (“static”) connectivity describes the spatial arrangement of the system components, which established the long-term potential for downstream transport^{4,5}. Functional (“dynamic”) connectivity is the interplay of spatial and temporal fluxes within the system for the short-term (i.e., storm event response)^{6,7}. The coevolution can be expressed as the initial landscape arrangement (structural) sets the general trends for hydrologic routing, but as hydrologic routing

causes the transfer of matter from hills to valleys (functional) and reorganizes the landscape over centuries and millennia, new flow paths are created and a new structural arrangement emerges⁸. This idea of hillslope connectivity has made it a focal point of recent research on quantifying landscape dynamics due to its potential for improved management of water and environmental systems⁹⁻¹¹. However, large-scale analysis of what controls structural connectivity has only recently been explored⁴ and climate change driven connectivity analyses have been limited to ecological connectivity^{12,13}.

For this study, we examine structural connectivity and climate due to the computational requirements to perform a large-scale functional connectivity analysis. We adopt a framework proposed by Ref¹⁴ in which we focus on long-term catchment response to climate in regard to structural connectivity as a first step in modeling of global hillslope connectivity. Before we can understand the role of climate change on structural hillslope connectivity, we need to first understand and model the drivers of these pathways. To explain, structural connectivity is informed by topographic features (e.g., elevation, slope, and roughness)⁵ and these features record the history of the landscape as they are the result of tectonic and climatic processes¹⁵. Furthermore, processes such as tectonics⁴ have been shown to play a crucial role on structural connectivity across large spatial domains. Despite these advances, we only have local (i.e., individual basins^{6,7,9}) or regional^{4,16,17} information about what drives connectivity, and we are lacking a global view of this phenomenon. Moreover, a model that allows capturing tectonic and climatic drivers worldwide is still missing, hindering our capability of making statements about future changes in hillslope connectivity.”

Furthermore, we add the following discussion to the conclusion section in lines 255 to 271:

“..... However, the time scale at which climate change occurs might be too short to manifest itself in structural connectivity changes for the other regions of the world but functional connectivity could be altered due to climate change as it is driven by climatic variables such as precipitation at short temporal scales^{6,7,16}. Furthermore, in regions where structural connectivity changes in a similar magnitude functional connectivity could be altered in completely different ways and so our takeaways should not be extrapolated to short-term durations. Future works to model functional connectivity at similar scales should be conducted to expand upon our efforts and capture a complete picture of hillslope connectivity as structural connectivity alone is not enough (see Ref³⁰). Our study presents an initial step in identifying regions across the world where structural hillslope connectivity dominates and is projected to be impacted by climate change. We encourage stakeholders to utilize our findings to focus initial climate planning efforts related to large-scale watershed management practices.”

I have added more thoughts, comments and suggestions in the annotated PDF attached to this review; I hope that the comments help to understand my thoughts and concerns.

R: Below we have listed verbatim with the line number the specific comments provided in the annotated PDF. Below each comment we have answered what has been completed to address the comment.

Line 15-18: At first sight, I cannot think of a way structural connectivity could be altered by climate change. Climate provides the hydrometeorological forcing of sediment mobilisation/erosion and transport, and hence is a major driver of functional connectivity. On the long term though, the activity of geomorphic processes (both exogenic and endogenic i.e. tectonic) may alter the configuration/topography of the landscape and, as a consequence, change structural connectivity. These explanations are based on a commonly accepted definition of structural vs. functional (sediment) connectivity, for example cited from Turnbull et al.:

Approaches to the study of connectivity within complex systems have often addressed structure (network architecture) and function (dynamical processes) separately. Structural connectivity (SC) measures of a system are used to quantify the level of configuration or arrangement of a network, whilst the functional connectivity (FC) of a system describes dynamical processes operating within a structurally connected network. SC thus derives from the system's anatomy, whereas FC is inferred from the system's process dynamics which are represented by fluxes and transformations of energy, matter or information between structural units.

Turnbull L, Hütt M-T, Ioannides AA, Kininmonth S, Poepl R, Tockner K, Bracken LJ, Keesstra S, Liu L, Masselink R, Parsons AJ. 2018. Connectivity and complex systems: Learning from a multi-disciplinary perspective. *Applied Network Science* 3(1): 47.

R: We address this concern earlier. We now make clearer the distinction of our definitions for structural and functional connectivity. We politely disagree that structural connectivity is only (or primarily) a function of tectonics – as other authors have shown the influence of long-term climatic changes to landscape development (see Ref²⁴) and hence alterations to structural connectivity over time.

In short, the repeated effects of functional connectivity can be integrated over long-enough periods to impact structural connectivity. We take great care regarding this distinction early on in the Introduction as well as later in the Discussion.

The following has been added at lines 54 to 87:

“ Hillslope connectivity refers to the linkage of upstream sources and downstream transport pathways (e.g., rills, gullies, rivers) and is informed by topographic features, which themselves encode the tectonic and climatic history of a landscape^{1,2}. Specifically, hillslope connectivity is composed of two interconnected types of connectivity, structural and functional³. Structural (“static”) connectivity describes the spatial arrangement of the system components, which established the long-term potential for downstream transport^{4,5}. Functional (“dynamic”) connectivity is the interplay of spatial and temporal fluxes within the system for the short-term (i.e., storm event response)^{6,7}. The coevolution can be expressed as the initial landscape arrangement (structural) sets the general trends for hydrologic routing, but as hydrologic routing causes the transfer of matter from hills to valleys (functional) and reorganizes the landscape over centuries and millennia, new flow paths are created and a new structural arrangement emerges⁸. This idea of hillslope connectivity has made it a focal point of recent research on quantifying

landscape dynamics due to its potential for improved management of water and environmental systems⁹⁻¹¹. However, large-scale analysis of what controls structural connectivity has only recently been explored⁴ and climate change driven connectivity analyses have been limited to ecological connectivity^{12,13}.

For this study, we examine structural connectivity and climate due to the computational requirements to perform a large-scale functional connectivity analysis. We adopt a framework proposed by Ref¹⁴ in which we focus on long-term catchment response to climate in regard to structural connectivity as a first step in modeling of global hillslope connectivity. Before we can understand the role of climate change on structural hillslope connectivity, we need to first understand and model the drivers of these pathways. To explain, structural connectivity is informed by topographic features (e.g., elevation, slope, and roughness)⁵ and these features record the history of the landscape as they are the result of tectonic and climatic processes¹⁵. Furthermore, processes such as tectonics⁴ have been shown to play a crucial role on structural connectivity across large spatial domains. Despite these advances, we only have local (i.e., individual basins^{6,7,9}) or regional^{4,16,17} information about what drives connectivity, and we are lacking a global view of this phenomenon. Moreover, a model that allows capturing tectonic and climatic drivers worldwide is still missing, hindering our capability of making statements about future changes in hillslope connectivity.”

Line 41-43: In my view, the focus on landscape dynamics, e.g. processes of erosion and sediment transfer, in line 41 clearly refers to functional connectivity (the term being avoided), while the very next sentence clearly refers to structural connectivity. To me, this is confusing.

R: Thank you for the comment. We have added the individual descriptions of structural and functional connectivity to this section. This has been addressed in lines 54 to 87 as previously provided.

Line 46-47:

refers to a term/sentence 7 lines further up - pls specify more clearly...

”...and model the drivers of sediment pathways” (?)

(1) While "tectonics" could in fact summarise (endogenic) "processes", "channel geomorphology" includes channel structure and material, and buffers/natural dams are landforms, not processes.

(2) Again, while processes drive functional connectivity, aspects of "channel geomorphology" and the existence and configuration of buffers (and barriers and blankets - I'd recommend to stick to Fryirs' terminology here) are factors of structural connectivity.

I think it is very important to distinguish the two aspects of (sediment) connectivity much more clearly in this paper. The same is true for the definition of what is a process, and what is a property, or landscape feature (e.g. a landform), or a material.

R: Thank you for the suggestion. We have updated this statement with a clearer definition of processes and removed the landforms aspect to avoid confusion. This has been addressed in lines 54 to 87 as previously provided.

Line 53: this, in my view, refers to both SC and FC. SC factors are static on the short-to-medium time scale, while FC factors are dynamic (=>forcing for sediment erosion and transport)

R: To make our manuscript clearer, we have added “structural” in front of the connectivity terms to make it clear this is the part on which we are focusing.

Line 54: This, in turn, can only refer to FC in my view (see previous comment).

R: We respectfully disagree as much research has highlighted the ability of long-term climatic variables such precipitation, evapotranspiration, and aridity to exert influence on landscape formation³¹⁻³⁷. We do however agree that functional connectivity dominates at short time-scales while structural connectivity exerts more control at long time-scales. Our study is predicting changes over 70 years, which we believe will be reflective of structural connectivity changes.

Line 58: clearly a SC index, so only addressing part of what is (in my opinion) "hillslope connectedness"

R: We have clarified that we are only looking at structural hillslope connectivity.

Specifically, we have added the following at line 74:

“For this study, we examine structural connectivity and climate due to the computational requirements to perform a large-scale functional connectivity analysis. We adopt a framework proposed by Ref¹⁴ in which we focus on long-term catchment response to climate in regard to structural connectivity as a first step in modeling of global hillslope connectivity.”

Line 59: As it stands here, it does not seem to address functional connectivity.

Do you really hypothesise that there was a statistical correlation between climate/tectonics and IC values ? On the long run, process shapes landscapes, right; and their topography is the most important factor of SC, and hence also the most important ingredient of IC. In this way, tectonics and climate could somehow influence the SC index value at very long time scales.

But is that really the aim of the study ?

I understood the aim is potential consequences of climate change on connectivity, which first and foremost affects FC. The time scale at which FC influences SC by changing landforms

is supposedly much longer than the time scale at which SC AND changing forcing influence FC.

R: Climate change will certainly impact functional connectivity, but also structural connectivity. At the present, the resolution of our numerical climate forecasting models is inadequate to assess short-term FC changes. However, we do believe that in the long-term, climate can impact structural connectivity (on average <2% - not a drastic change but potentially an important one). We hope that the research community can one day develop the necessary tools for us to be able to resolve functional connectivity over the course of centuries, but at current we are limited to assessing structural connectivity impacts.

Line 78-80: The point is, why is there a correlation.

My explanation would be: IC is high in mountains due to steepness; and precipitation is generally higher in mountains because topography causes higher precipitation. The correlation of IC and precipitation would then be a coincidence rather than something that could explain a causative relationship.

Moreover, we need to consider the quality of the Hydrosheds river dataset in arid regions. This dataset uses a constant threshold of 100 cells for channel initiation, which means that channels might be modelled where in reality there is none. Due to the channel dataset relying exclusively on topography, the influence of climate on the density of the channel network is not considered, and hence IC values might be overestimated (with non-existent rivers as the target).

The influence of "real" channel density on connectivity is discussed (and implemented via a dataset containing the "real" river network in comparison with that modelled on the basis of a DEM only):

Gay A, Cerdan O, Mardhel V, Desmet M. 2016. Application of an index of sediment connectivity in a lowland area. *Journal of Soils and Sediments* 16(1): 280–293.

This more recent paper seems to provide a global river network dataset that takes variable drainage density into account:

Lin P, Pan M, Wood EF, Yamazaki D, Allen GH. 2021. A new vector-based global river network dataset accounting for variable drainage density. *Scientific data* 8(1): 28.

R: Please refer to the comment addressing the issue between elevation and climate drivers (see also Figures R1-R2) at the beginning of our responses. We have added notes of caution within this section as well as discussed it in the limitations.

Line 80-81: if the results suggest a positive relationship between IC and precipitation, while the US results suggest a negative one, how can this difference be "slight" ??

R: Thank you for the comment. We agree and have removed the word "slightly."

Line 86: typo?

R: Thank you for the comment. We have removed the word “altered.”

Line 92-93: A correlation of -0,4 suggests a negative correlation, not a "weaker positive correlation". The observation is highly plausible because longer rivers imply longer pathways and hence lower IC values.

R: Thank you for the comment. The word positive should not have been there and we have removed it in the revised manuscript.

Line 132-133: In my view, they cannot be for conceptual reasons, see the definition of structural connectivity! Conversely, atmospheric changes/changes in hydrometeorological forcing are supposedly directly translated into changes in FC !

R: We agree with this, and we were trying to support our modeling results. Based on comments for this paragraph from Reviewer 3, we have decided to remove the entire paragraph from Line 127-133 of the original manuscript.

Line 148-151: See this reference that investigated changes in both structural and functional connectivity with time using data on stream flow, sediment flux, precipitation and multitemporal DEMs:

Lane SN, Bakker M, Gabbud C, Micheletti N, Saugy J-N. 2017. Sediment export, transient landscape response and catchment-scale connectivity following rapid climate warming and Alpine glacier recession. *Geomorphology* 277: 210–227.

R: Thank you for this suggested study. We have added this reference to the section with a new sentence.

The following has been added at lines 214 to 242:

“... Analysis of observational data conducted by Ref¹⁹ supports this concept as the authors found that glacier recession leads to increased connectivity between the upper basin and proglacial areas but river reworking of glacial till and coarse sediment create negative feedbacks that reduce sediment export.”

Line 157: In my view, such a statement is difficult. If your analysis is right, then the IC values change. IC, however, represents a very abstract phenomenon and is NOT readily translated in, for example, sediment delivery ratios (as a measure of functional connectivity), especially considering the global scale of your study. Even if the regression analyses done here were correct in predicting a change in topography that leads to a different index value, I still expect huge differences in FC among areas that have essentially the same SC.

R: We agree with the Reviewer that two basins with the same SC can have different FC responses. However, it is still a worthy goal to estimate the SC for the basins as it provides basic information that can aid in understanding transport dynamics. As mentioned earlier, we are presently limited by computational techniques to simulate the coupled effects of FC and SC in basins, thus we must provide what information we can, which relates to SC changes that may have variable influence to FC responses. We now discuss the Reviewer's good point regarding how SC is not the lone predictor of catchment response in the discussion.

The following has been added at lines 255 to 263:

“..... However, the time scale at which climate change occurs might be too short to manifest itself in structural connectivity changes for the other regions of the world but functional connectivity could be altered due to climate change as it is driven by climatic variables such as precipitation at short temporal scales^{6,7,16}. Furthermore, in regions where structural connectivity changes in a similar magnitude functional connectivity could be altered in completely different ways and so our takeaways should not be extrapolated to short-term durations. Future works to model functional connectivity at similar scales should be conducted to expand upon our efforts and capture a complete picture of hillslope connectivity as structural connectivity alone is not enough (see Ref³⁰).

Line 158-161: I agree that connectivity hotspots should be looked at, because changes in geomorphic activity and hence functional connectivity will lead to efficient propagation of changes through catchments, as suggested above (water quality issues as more sediment is being delivered to channels). However, your study does not focus on FC changes in areas of different (especially high) SC, it focuses entirely on SC changes, which, according to the IC index used, reacts almost exclusively to changes in topography - which, in turn, takes place on longer time scales, while the changes in forcing in terms of climate change affect FC much earlier, and this might be stronger where SC is high.

R: We addressed in an earlier comment how the different timescales can indeed impact one another even on timescales of 10 to 100 years (see Ref⁸).

Line 164-165: My point is that, if a climate-driven numerical model is capable of correctly predicting (changes in) water and sediment fluxes, FC will "automatically" emerge from model results, and SC indices might not be necessary in this case.

This study shows that model results vastly differ, but it reports that "Functional connectivity (rainfall forcing) [is] more important than structural connectivity":

Baartman JE, Nunes JP, Masselink R, Darboux F, Biellers C, Degre A, Cantreul V, Cerdan O, Grangeon T, Fiener P, Wilken F, Schindewolf M, Wainwright J. 2020. What do models tell us about water and sediment connectivity? *Geomorphology*: 107300.

R: We would agree that rainfall is more important for functional connectivity but also feel it has importance for structural connectivity. The short-term response of a catchment to rainfall is functional connectivity. The long-term response of a catchment to rainfall is structural connectivity. We see this framework represented in Ref¹⁴ and adapt it for our study.

Regarding the Baartman et al., 2020 (Ref³⁰) study, we agree with the authors that functional responses are more important than structural connectivity. However, as mentioned in our earlier comment, our computational abilities are only capable of resolving the structural response. We aim to resolve functional responses in the future. We reference the Baartman et al. study and explicitly discuss that our model considers structural connectivity, which is only one component, and that functional connectivity may emerge as a greater control and that the modeling community should seek to resolve those dynamics in the future.

This response can be found in Lines 261 to 271:

“Future works to model functional connectivity at similar scales should be conducted to expand upon our efforts and capture a complete picture of hillslope connectivity as structural connectivity alone is not enough (see Ref³⁰). Our study presents an initial step in identifying regions across the world where structural hillslope connectivity dominates and is projected to be impacted by climate change.”

Line 168: This is not detailed enough in my opinion. I looked it up in the cited reference, but the latter does not mention how the W component of IC is computed. While at first I assumed you were taking the USLE C factor (as in Borselli et al., to whom you refer in Ref 14), but upon more careful reading of Ref14 you mention "microtopographic undulations", and I realised that I had to look up supplementary information in order to be able to fully reproduce the methods (which is suboptimal in my opinion). It turns out that the "microtopographic undulation" is the roughness index introduced by Cavalli et al. 2013. While the latter study computes this roughness/impedance index on a comparatively high-resolution DEM (where it really represents something like "microtopography"), Ref14 uses a DEM resolution of 10m, which is already questionable regarding "microtopography". In this manuscript, you use a 90 m DEM, and this is definitely not able to represent "microtopographic undulations" any more - considering that the roughness index is computed on a 5x5 neighbourhood (corresponding to a square of 450x450 meters) !

While I acknowledge that DEM data at the required resolution are not available at the global scale, and that the computational cost would be excessive, I still question the applicability of the roughness index at that coarse resolution and would recommend using C factor based on globally available landcover data.

Fortunately you mention the "stream target" here (which is not the case in Ref14, at least no mention in the methods section).

R: Thank you for the comment. We have added detailed descriptions of the methodology in the Methods section with all equations as well as expanded information on the statistical model. In regards to the C factor, we have made sure to add this as a future step and acknowledged the limitations of our current approach.

The following has been added at lines 272 to 400 (Methods):

“To represent structural hillslope connectivity across the globe, we calculated the Index of Connectivity (IC) based on the same methodology as Ref⁴, which describes the probability of

sediment from an upslope point to travel to a downslope target (streams). IC is empirically defined as:

$$IC = \log_{10}\left(\frac{D_{up}}{D_{dn}}\right) \quad (1)$$

where D_{up} and D_{dn} represent the upslope and downslope elements of connectivity. IC values can range from $[-\infty, \infty]$ with greater values indicating higher connectivity.

The upslope component, D_{up} , represents the potential of sediment yields from upslope sources to be routed downward and is defined as:

$$D_{up} = \bar{W}\bar{S}\sqrt{A} \quad (2)$$

where \bar{W} is the average weighting factor of the contributing upslope area, \bar{S} is the average slope of the contributing upslope area (m/m) and A is the contributing upslope area (m²).

Next, we define the downslope component, D_{dn} , as the probability of the sediment flow to travel along the flow path arriving to the nearest target. The downslope component is defined as:

$$D_{dn} = \sum_i \frac{d_i}{W_i S_i} \quad (3)$$

where d_i is the flow path length to the downstream channel for the i^{th} cell at the steepest slope direction (m). W_i and S_i are the weighting factor and slope gradient, respectively, at the i^{th} cell.

For our analysis we determine the weighting factor, W , based on the roughness index (RI) or surface roughness as defined by Ref⁹. RI is the standard deviation of the residual topography computed over a 5×5 cell moving window defined as:

$$RI = \sqrt{\frac{\sum_{i=1}^{25} (x_i - x_m)^2}{25}} \quad (4)$$

where x_i is the value of residual topography at the i^{th} cell within the window and x_m is the mean of the cells within the moving window. Finally, the weighting factor is defined as:

$$W = 1 - \left(\frac{RI}{RI_{max}}\right) \quad (5)$$

where RI_{max} is the maximum RI value for a region.

For inputs to calculate IC, we utilized DEMs with a 90-meter spatial resolution from HydroSHEDS²⁰. To determine d_i in Eq. 3, we defined targets as nearby streams. We utilized streams from the HydroRIVERS dataset²¹. Additionally, HydroSHEDS' 90-m DEM does not provide coverage above 60° N and so we do not conduct our analysis for Arctic, Greenland, Iceland, Scandinavia, and Siberia. For the primary analysis we utilized HydroBASINS²¹ level 5 as the unit to conduct basin-averaged analyses. More specifically, for IC and the drivers we take the average value across the level 5 defined basin and ran our analyses.

To examine the drivers of connectivity at a global level and examine its sensitivity to climate change, we built a regression model based on the Generalized Additive Model for Location, Scale and Shape (GAMLSS)²². We utilized tectonic and climate drivers from Refs^{23,24} as they computed the basin averages at the same HydroBASINS level. The tectonic proxies consist of the Peak Ground Acceleration (PGA) from the Global Earthquake Model GEM²⁵, mean river segment slope, river segment length, and mean elevation of river profile segment. For climatic drivers, we utilized precipitation and evapotranspiration from the WorldClim dataset²⁶. We examined the correlation between the drivers (predictors) using a correlogram (Fig. S1) and found weak dependence among these covariates. Other predictors of river concavity, total relief, and

aridity index from the Ref ^{23,24} were excluded in our initial data analysis because of strong correlation with other covariates (i.e., multicollinearity). Next, we applied stepwise model selection to choose the best model for our purposes using the Schwarz Bayesian Criterion (SBC)²⁷ as selection criterion. We examined multiple distributions (Table S1) and model configurations to select the best model based on SBC.

For this study we utilize the 4-parameter Skew t type I (ST1) distribution with the parameters depending on the predictors. The probability distribution function (pdf) for the ST1 is given as:

$$f_Y(y | \mu, \sigma, v, \tau) = \begin{cases} \frac{c}{\sigma_0} \left[1 + \frac{v^2 z^2}{\tau} \right]^{-(r+1)/2} & \text{if } y < \mu_0 \\ \frac{c}{\sigma_0} \left[1 + \frac{z^2}{v^2 \tau} \right]^{-(r+1)/2} & \text{if } y \geq \mu_0 \end{cases} \quad (6)$$

where μ is the location shift parameter $[-\infty, \infty]$, σ is the scaling parameter $[-\infty, \infty]$, v is the skewness parameter $[-\infty, \infty]$, τ is the kurtosis parameter $[0, \infty]$, and $z = (y - \mu)/\sigma$. For our analysis y is the basin averaged IC values across the globe $[-\infty, \infty]$. More information on the distribution can be found at Ref²². For each the distribution parameters the following linear equations where derived:

$$\mu = -5.092 - (9.037 \times 10^{-5})PET - (7.427 \times 10^{-1})PGA + (2.981 \times 10^{-4})P + (1.210 \times 10^2)S \quad (7)$$

$$\ln(\sigma) = -2.476 + (1.078 \times 10^{-4})P + (2.314 \times 10^{-5})PL + (7.380 \times 10^1)S \quad (8)$$

$$v = 1.486 - (2.410 \times 10^{-4})EL - (1.134 \times 10^{-3})PET + (2.768)PGA - (1.921 \times 10^{-5})PL - (6.396 \times 10^1)S \quad (9)$$

$$\ln(\tau) = -7.088 \times 10^{-1} + (7.614 \times 10^{-4})EL - (3.057)PGA + (6.678 \times 10^{-5})PL + (2.139 \times 10^2)S \quad (10)$$

where EL is elevation (m), PET is potential evapotranspiration (mm/yr.), PGA is peak ground acceleration (g), P is precipitation (mm/yr.), PL is profile length (m), and S is slope (-) as described above. For the ST1, μ and v have identity link functions whereas logarithmic link functions are used for σ and τ . The goodness of fit statistics for residuals consisting of mean, variance, coefficient of skewness, coefficient of kurtosis, and Filliben correlation have values of -0.043, 0.957, 0.029, 3.235, and 0.999, respectively. To calculate a mean IC value for a given basin, the basin average values for the inputs (e.g., PET , PGA) are plugged into eqs. 7-10 and the estimated parameters are utilized to compute the 50th percentile from the ST1 (eq. 6). These IC values are shown in Fig. 1.

To assess the impact of climate change we utilized precipitation and PET from 34 climate models part of the Coupled Model Intercomparison Project Phase 6 (CMIP6)³⁸ and computed a ratio of change between historical (1970-2000) and future periods (2070-2100). We then multiply the historical (1970-2000) annual average precipitation (Fig. S2) and PET (Fig. S3) for each basin by the ratio and recalculate IC based on our regression model (eqs. 6-10). We provide the ratios of change in Fig. S4 (precipitation) and Fig. S5 (PET). We chose these two predictors for our sensitivity analysis as the time scale of which changes in climate occur are much smaller compared to tectonic drivers.

The primary limitations of our study are related to the features in the model. The initial calculation of IC based on DEMs is dependent upon the provided stream targets and artifacts captured in the DEM. First, our IC estimates may be overestimated as the HydroRIVERS dataset uses a constant flow accumulation threshold of 100 cells where channels could be shown to exist that are not actually there. Future work should examine the influence of channel density on

connectivity similar to Ref³⁹ with newer river datasets (e.g., Ref⁴⁰). We utilized a roughness factor described by Ref⁹ for the weighting factor described above but this may not be adequate to capture topographic roughness when utilizing a 90-m DEM. For future studies, we suggest the exploration of utilizing the Universal Soil Loss Equation C-factor in place of RI as shown in Refs^{28,29}. This means that we did not capture the role of land use changes on structural connectivity in the future where transitions to urban or agricultural lands could offset changes due to climate. Further analyses of the impact of climate change should be performed regarding functional or dynamic connectivity, which utilizes physically-based models to understand hillslope response^{6,7}. Functional connectivity is driven by shorter precipitation events which are expected to change in magnitude and frequency in the future⁴¹. The use of physically-based models could also aid with making causal statements beyond the correlation results presented here. Finally, due to the larger time scales of tectonic actions, we did not explore the sensitivity of these parameters. However, future modeling efforts should explore these drivers to understand exactly the time scale at which structural connectivity is impacted. ”

Line 195-196: This is surely correct. BUT it is precisely the difference in time scales on which my criticism is based upon. Yes, climate change occurs quickly, but the reaction of a geomorphic system that would lead to a change in IC (=>SC) is MUCH too long compared to the immediate effect on FC (which is not considered in your study).

R: For our study the focus is on structural connectivity. We agree and have added statements discussing this in the conclusion of the revised manuscript.

The following has been added at lines 255 to 260:

“..... However, the time scale at which climate change occurs might be too short to manifest itself in structural connectivity changes for the other regions of the world but functional connectivity could be altered due to climate change as it is driven by climatic variables such as precipitation at short temporal scales^{6,7,16}. Furthermore, in regions where structural connectivity changes in a similar magnitude functional connectivity could be altered in completely different ways and so our takeaways should not be extrapolated to short-term durations.”

Line 200: Yes regarding hydrological connectivity, but not necessarily for sediment connectivity (in a widely sealed urban landscape)

R: We have changed the focus to be on the C factor as suggested and removed this statement.

Line 202-203: Is that true generally ? Can you provide a reference for this assumption ?

R: Thank you for the comment. We have changed the ideas within this paragraph and this statement has been removed.

Reviewer #3 (Remarks to the Author):

Review of “Climate change amplifies structural hillslope connectivity at the global scale” by Michalek et al.

This manuscript presents a new global (excluding high latitudes) dataset of connectivity indices and then explores the correlation of that connectivity index with various topographic, climatic, and seismic datasets. The manuscript then introduces a statistical model to predict connectivity index and attempts to forecast the impact of climate predictions (precipitation and evapotranspiration) on connectivity. The study finds that the connectivity index is more strongly correlated with topographic and seismic properties than with climate properties, but still shows some correlation with precipitation and evapotranspiration (significance was not evaluated). By applying forecasted precipitation and evapotranspiration, the study predicts that about half of all basin will have greater connectivity (and presumably half of all basins will have lower connectivity).

R: Thank you for all of the comments below. We have made sure to go through the manuscript and address all points necessary to improve its' clarity.

The topic is an interesting one to explore, but paper unfortunately does a very poor job of explaining what was actually done, making it impossible to evaluate the work or understand its implications. At the most fundamental level, it is not clear from this manuscript what is meant by “connectivity” or how it was measured. The first paragraph motivates the study by highlighting the importance of connectivity within the stream and river network, and the next paragraph moves into connectivity between hillslopes and the stream network. These paragraphs also discuss connectivity of water, sediment, and nutrients, which are all related but driven by different sets of processes. It’s not clear to me which form of connectivity is being evaluated here, and there is no explanation of how it is measured. How can the reader interpret a connectivity index of -4 without any explanation?

R: Thank you for the comment and we apologize for the limited details. We are examining structural connectivity to the nearest stream target and coupling the index of connectivity (IC) with an additive model to predict future IC values to quantify the impact of climate change. For IC values, the smaller or more negative a value means that the area is less connected across the basin. To explain in Fig. 1, -6 would be lowest connectivity value (IC) and -3 would be high the highest connectivity value (IC). To make this point clear we have rewritten multiple sections of the introduction and section discussing IC to bring clarity to the manuscript.

The following has been added and modified at lines 74 to 107 in the introduction:

“ For this study, we examine structural connectivity and climate due to the computational requirements to perform a large-scale functional connectivity analysis. We adopt a framework proposed by Ref22 in which we focus on long-term catchment response to climate in regard to structural connectivity as a first step in modeling of global hillslope connectivity. Before we can understand the role of climate change on structural hillslope connectivity, we need to first understand and model the drivers of these pathways. To explain, structural connectivity is informed by topographic features (e.g., elevation, slope, and roughness)¹⁴ and these features record the history of the landscape as they are the result of tectonic and climatic processes²³.

Furthermore, processes such as tectonics¹³ have been shown to play a crucial role on structural connectivity across large spatial domains. Despite these advances, we only have local (i.e., individual basins^{15,16,18}) or regional^{5,13,24} information about what drives connectivity, and we are lacking a global view of this phenomenon. Moreover, a model that allows capturing tectonic and climatic drivers worldwide is still missing, hindering our capability of making statements about future changes in hillslope connectivity.

In this work, we ask the questions: what are the drivers of global structural hillslope connectivity and how will climate change alter the connectedness of landscapes? We hypothesize that climate change will amplify connectivity where rainfall is projected to intensify and that it will dampen connectivity where future drought conditions are projected to prevail. We estimate structural hillslope connectivity for over 3,500 basins across the globe at a spatial resolution of 90 m. We quantify the potential strength of structural hillslope connectedness with the Index of Connectivity (IC)²⁵ and develop a statistical model to explain its climatic and tectonic drivers (details in Methods). We evaluate our hypothesis on the effects of climate change on connectivity by applying the model to four future climate scenarios.”

In the caption of Figure 1 we have added the following statement:

“To provide context, IC can range from $[-\infty, \infty]$ and the smaller (more negative) a value, the less connected a basin is. For this plot, low connectivity is on the darker end of the spectrum (-6) and higher connectivity is at the lighter end of the spectrum (-3). ”

Moreover, the finding that is claimed in the title is not properly evaluated in the manuscript. Based on my reading of the work, about half of all basins are predicted to have greater connectivity and about half of all basins are expected to have lower connectivity under various climate scenarios. In other words, the impact of climate change on connectivity is a wash. I may be reading that incorrectly, but if so, the paper needs to make that clear, starting with a proper evaluation of the significance of predicted impact of climate change on connectivity.

R: Thank you for the comment. The goal of the paper is to show that structural hillslope connectivity is projected to change due to climate change in some parts of the world. This can be either an increase or decrease in connectivity. Your initial reading is correct. We have modified the title to reflect the content of the work.

We have made the points clearer in the abstract at the lines 22 to 26:

“We find climate change is projected to alter global-scale connectivity at the end of the century (2070 to 2100) by up to 4% for increasing greenhouse gas emission scenarios. Notably, the Ganges River, the world’s most populated river basin, is projected to experience a drastic increase in connectivity. Conversely, the Amazon River and the Pacific coast of Patagonia are projected to experience the largest decreases in connectivity.”

Additionally, we have modified the conclusion in lines 255 to 271 to reflect the overall impact we find:

“..... However, the time scale at which climate change occurs might be too short to manifest itself in structural connectivity changes for the other regions of the world but functional connectivity could be altered due to climate change as it is driven by climatic variables such as precipitation at short temporal scales^{6,7,16}. Furthermore, in regions where structural connectivity changes in a similar magnitude functional connectivity could be altered in completely different ways and so our takeaways should not be extrapolated to short-term durations. Future works to model functional connectivity at similar scales should be conducted to expand upon our efforts and capture a complete picture of hillslope connectivity as structural connectivity alone is not enough (see Ref³⁰). Our study presents an initial step in identifying regions across the world where structural hillslope connectivity dominates and is projected to be impacted by climate change. We encourage stakeholders to utilize our findings to focus initial climate planning efforts related to large-scale watershed management practices.”

Also, without knowing how “structural connectivity” is calculated, I’m left wondering how flexible it is to changes in climate. The “structural” part of it makes me think that it’s a topographic measure. If that’s true, the impact of climate on structural connectivity would require topography to adjust to the new climatic regime before those connectivity adjustments are realized.

R: Thank you for the comment. We have added an in-depth description of the calculations to the Methods section. The IC metric is a DEM derived index that we couple with a statistical model to understand the impact of climate on structural connectivity.

The following has been added at lines 272 to 400 (Methods):

“To represent structural hillslope connectivity across the globe, we calculated the Index of Connectivity (IC) based on the same methodology as Ref⁴, which describes the probability of sediment from an upslope point to travel to a downslope target (streams). IC is empirically defined as:

$$IC = \log_{10} \left(\frac{D_{up}}{D_{dn}} \right) \quad (1)$$

where D_{up} and D_{dn} represent the upslope and downslope elements of connectivity. IC values can range from $[-\infty, \infty]$ with greater values indicating higher connectivity.

The upslope component, D_{up} , represents the potential of sediment yields from upslope sources to be routed downward and is defined as:

$$D_{up} = \bar{W} \bar{S} \sqrt{A} \quad (2)$$

where \bar{W} is the average weighting factor of the contributing upslope area, \bar{S} is the average slope of the contributing upslope area (m/m) and A is the contributing upslope area (m^2).

Next, we define the downslope component, D_{dn} , as the probability of the sediment flow to travel along the flow path arriving to the nearest target. The downslope component is defined as:

$$D_{dn} = \sum_i \frac{d_i}{W_i S_i} \quad (3)$$

where d_i is the flow path length to the downstream channel for the i^{th} cell at the steepest slope direction (m). W_i and S_i are the weighting factor and slope gradient, respectively, at the i^{th} cell.

For our analysis we determine the weighting factor, W , based on the roughness index (RI) or surface roughness as defined by Ref⁹. RI is the standard deviation of the residual topography computed over a 5×5 cell moving window defined as:

$$RI = \sqrt{\frac{\sum_{i=1}^{25} (x_i - x_m)^2}{25}} \quad (4)$$

where x_i is the value of residual topography at the i^{th} cell within the window and x_m is the mean of the cells within the moving window. Finally, the weighting factor is defined as:

$$W = 1 - \left(\frac{RI}{RI_{max}} \right) \quad (5)$$

where RI_{max} is the maximum RI value for a region.

For inputs to calculate IC, we utilized DEMs with a 90-meter spatial resolution from HydroSHEDS²⁰. To determine d_i in Eq. 3, we defined targets as nearby streams. We utilized streams from the HydroRIVERS dataset²¹. Additionally, HydroSHEDS' 90-m DEM does not provide coverage above 60° N and so we do not conduct our analysis for Arctic, Greenland, Iceland, Scandinavia, and Siberia. For the primary analysis we utilized HydroBASINS²¹ level 5 as the unit to conduct basin-averaged analyses. More specifically, for IC and the drivers we take the average value across the level 5 defined basin and ran our analyses.

To examine the drivers of connectivity at a global level and examine its sensitivity to climate change, we built a regression model based on the Generalized Additive Model for Location, Scale and Shape (GAMLSS)²². We utilized tectonic and climate drivers from Refs^{23,24} as they computed the basin averages at the same HydroBASINS level. The tectonic proxies consist of the Peak Ground Acceleration (PGA) from the Global Earthquake Model GEM²⁵, mean river segment slope, river segment length, and mean elevation of river profile segment. For climatic drivers, we utilized precipitation and evapotranspiration from the WorldClim dataset²⁶. We examined the correlation between the drivers (predictors) using a correlogram (Fig. S1) and found weak dependence among these covariates. Other predictors of river concavity, total relief, and aridity index from the Ref^{23,24} were excluded in our initial data analysis because of strong correlation with other covariates (i.e., multicollinearity). Next, we applied stepwise model selection to choose the best model for our purposes using the Schwarz Bayesian Criterion (SBC)²⁷ as selection criterion. We examined multiple distributions (Table S1) and model configurations to select the best model based on SBC.

For this study we utilize the 4-parameter Skew t type I (ST1) distribution with the parameters depending on the predictors. The probability distribution function (pdf) for the ST1 is given as:

$$f_Y(y | \mu, \sigma, v, \tau) = \begin{cases} \frac{c}{\sigma_0} \left[1 + \frac{v^2 z^2}{\tau} \right]^{-(r+1)/2} & \text{if } y < \mu_0 \\ \frac{c}{\sigma_0} \left[1 + \frac{z^2}{v^2 \tau} \right]^{-(r+1)/2} & \text{if } y \geq \mu_0 \end{cases} \quad (6)$$

where μ is the location shift parameter $[-\infty, \infty]$, σ is the scaling parameter $[-\infty, \infty]$, v is the skewness parameter $[-\infty, \infty]$, τ is the kurtosis parameter $[0, \infty]$, and $z = (y - \mu)/\sigma$. For our analysis y is the

basin averaged IC values across the globe $[-\infty, \infty]$. More information on the distribution can be found at Ref²². For each the distribution parameters the following linear equations were derived:

$$\mu = -5.092 - (9.037 \times 10^{-5})PET - (7.427 \times 10^{-1})PGA + (2.981 \times 10^{-4})P + (1.210 \times 10^2)S \quad (7)$$

$$\ln(\sigma) = -2.476 + (1.078 \times 10^{-4})P + (2.314 \times 10^{-5})PL + (7.380 \times 10^1)S \quad (8)$$

$$v = 1.486 - (2.410 \times 10^{-4})EL - (1.134 \times 10^{-3})PET + (2.768)PGA - (1.921 \times 10^{-5})PL - (6.396 \times 10^1)S \quad (9)$$

$$\ln(\tau) = -7.088 \times 10^{-1} + (7.614 \times 10^{-4})EL - (3.057)PGA + (6.678 \times 10^{-5})PL + (2.139 \times 10^2)S \quad (10)$$

where EL is elevation (m), PET is potential evapotranspiration (mm/yr.), PGA is peak ground acceleration (g), P is precipitation (mm/yr.), PL is profile length (m), and S is slope (-) as described above. For the ST1, μ and v have identity link functions whereas logarithmic link functions are used for σ and τ . The goodness of fit statistics for residuals consisting of mean, variance, coefficient of skewness, coefficient of kurtosis, and Filliben correlation have values of -0.043, 0.957, 0.029, 3.235, and 0.999, respectively. To calculate a mean IC value for a given basin, the basin average values for the inputs (e.g., PET , PGA) are plugged into eqs. 7-10 and the estimated parameters are utilized to compute the 50th percentile from the ST1 (eq. 6). These IC values are shown in Fig. 1.

To assess the impact of climate change we utilized precipitation and PET from 34 climate models part of the Coupled Model Intercomparison Project Phase 6 (CMIP6)³⁸ and computed a ratio of change between historical (1970-2000) and future periods (2070-2100). We then multiply the historical (1970-2000) annual average precipitation (Fig. S2) and PET (Fig. S3) for each basin by the ratio and recalculate IC based on our regression model (eqs. 6-10). We provide the ratios of change in Fig. S4 (precipitation) and Fig. S5 (PET). We chose these two predictors for our sensitivity analysis as the time scale of which changes in climate occur are much smaller compared to tectonic drivers.

The primary limitations of our study are related to the features in the model. The initial calculation of IC based on DEMs is dependent upon the provided stream targets and artifacts captured in the DEM. First, our IC estimates may be overestimated as the HydroRIVERS dataset uses a constant flow accumulation threshold of 100 cells where channels could be shown to exist that are not actually there. Future work should examine the influence of channel density on connectivity similar to Ref³⁹ with newer river datasets (e.g., Ref⁴⁰). We utilized a roughness factor described by Ref⁹ for the weighting factor described above but this may not be adequate to capture topographic roughness when utilizing a 90-m DEM. For future studies, we suggest the exploration of utilizing the Universal Soil Loss Equation C-factor in place of RI as shown in Refs^{28,29}. This means that we did not capture the role of land use changes on structural connectivity in the future where transitions to urban or agricultural lands could offset changes due to climate. Further analyses of the impact of climate change should be performed regarding functional or dynamic connectivity, which utilizes physically-based models to understand hillslope response^{6,7}. Functional connectivity is driven by shorter precipitation events which are expected to change in magnitude and frequency in the future⁴¹. The use of physically-based models could also aid with making causal statements beyond the correlation results presented here. Finally, due to the larger time scales of tectonic actions, we did not explore the sensitivity of these parameters. However, future modeling efforts should explore these drivers to understand exactly the time scale at which structural connectivity is impacted. ”

I believe that there could be some interesting findings here but the presentation needs a substantial amount of work so that the reader can understand what’s going on.

Comments by line number

70. I assume “changes in elevation” refers to “topographic relief” and not temporal evolution

R: Thank you for the comment. This is correct and we have made this change.

83. “importance of climatic variables increases” with respect to what?

R: Thank you for the comment. This is in regard to the CONUS study. We have clarified this statement within the manuscript. These changes can be found in lines 136 to 140 in the revised manuscript.

85. “longer-term climate variables” What are these? Were they evaluated?

R: Thank you for the comment. We meant for this statement to indicate the variables of precipitation and PET previously discussed in the paragraph. We have updated this sentence to make this clearer to the reader.

89. “drivers” should be “proxies”

R: Thank you for the comment. We have made this change.

89. “river profile slope” over which spatial scale and on which segments?

R: Thank you for the comment. For our analysis we utilized the river profile slope from the Ref²³ and available from the GloPro database (<https://doi.org/10.17636/01058162>). Their study describes the methodology in depth. For the river profile slope, the authors utilized LSDTopoTools with a 30-m DEM resolution and a 25 sq. km flow accumulation threshold to determine the profile slope and elevation for 333,502 river longitudinal profiles across the globe. The values we utilized were the basin averaged as taken from Ref²⁴. We have added the reference to this line.

92. “river profile elevation” Elevation of which part of the river?

R: Thank you for the comment. The river profile elevation or relief is along all channels utilized in Ref²³. Please see the previous comment on where to find detailed information on their

methodology. We have added the references to this section so that the reader can clearly find the study where the data are from.

113 - . Good to state in the main text which model parameters you're tweaking from the climate model and not require the reader to go to the methods.

R: Thank you for the comment. We have added a sentence specifying what the parameters are.

The following has been added at lines 180: "Specifically, we focus on the impact of PET and precipitation to changes in connectivity."

127 – 133. I don't understand what is trying to be said here. It's not clear from the manuscript how to intuitively relate changes in climate to connectivity.

R: Thank you for pointing out the miscommunication. For this paragraph, we are trying to communicate that the statistical model (gamlss) utilized with the topographic model (IC) is not linear and so if we change the inputs of precipitation or PET in the future for a given basin by say 5%, then the IC predicted value will not change by 5%. To avoid confusion for this point, we have decided to omit this paragraph.

Reference

*Note references are for both manuscript text and responses in order in which they appear within this document.

- 1 Cao, W. *et al.* Palaeolatitudinal distribution of lithologic indicators of climate in a palaeogeographic framework. *Geological Magazine* **156**, 331-354 (2018). <https://doi.org/10.1017/s0016756818000110>
- 2 Champagnac, J.-D., Molnar, P., Sue, C. & Herman, F. Tectonics, climate, and mountain topography. *Journal of Geophysical Research: Solid Earth* **117**, n/a-n/a (2012). <https://doi.org/10.1029/2011jb008348>
- 3 Turnbull, L. *et al.* Connectivity and complex systems: learning from a multi-disciplinary perspective. *Appl Netw Sci* **3**, 11 (2018). <https://doi.org/10.1007/s41109-018-0067-2>
- 4 Husic, A. & Michalek, A. Structural Hillslope Connectivity Is Driven by Tectonics More Than Climate and Modulates Hydrologic Extremes and Benefits. *Geophysical Research Letters* **49** (2022). <https://doi.org/10.1029/2022gl099898>
- 5 Heckmann, T. *et al.* Indices of sediment connectivity: opportunities, challenges and limitations. *Earth-Science Reviews* **187**, 77-108 (2018). <https://doi.org/10.1016/j.earscirev.2018.08.004>
- 6 Mahoney, D. T., Fox, J., Al-Aamery, N. & Clare, E. Integrating connectivity theory within watershed modelling part II: Application and evaluating structural and functional connectivity. *Sci Total Environ* **740**, 140386 (2020). <https://doi.org/10.1016/j.scitotenv.2020.140386>
- 7 Mahoney, D. T., Fox, J., Al-Aamery, N. & Clare, E. Integrating connectivity theory within watershed modelling part I: Model formulation and investigating the timing of sediment connectivity. *Sci Total Environ* **740**, 140385 (2020). <https://doi.org/10.1016/j.scitotenv.2020.140385>
- 8 Litwin, D. G., Tucker, G. E., Barnhart, K. R. & Harman, C. J. Groundwater Affects the Geomorphic and Hydrologic Properties of Coevolved Landscapes. *J Geophys Res-Earth* **127** (2022). <https://doi.org/10.1029/2021JF006239>
- 9 Cavalli, M., Trevisani, S., Comiti, F. & Marchi, L. Geomorphometric assessment of spatial sediment connectivity in small Alpine catchments. *Geomorphology* **188**, 31-41 (2013). <https://doi.org/10.1016/j.geomorph.2012.05.007>
- 10 Heckmann, T. & Vericat, D. Computing spatially distributed sediment delivery ratios: inferring functional sediment connectivity from repeat high-resolution digital elevation models. *Earth Surface Processes and Landforms* **43**, 1547-1554 (2018). <https://doi.org/10.1002/esp.4334>
- 11 Najafi, S., Dragovich, D., Heckmann, T. & Sadeghi, S. H. Sediment connectivity concepts and approaches. *Catena* **196** (2021). <https://doi.org/10.1016/j.catena.2020.104880>
- 12 Krosby, M., Tewksbury, J., Haddad, N. M. & Hoekstra, J. Ecological connectivity for a changing climate. *Conserv Biol* **24**, 1686-1689 (2010). <https://doi.org/10.1111/j.1523-1739.2010.01585.x>
- 13 Murphy, E. J. *et al.* Global Connectivity of Southern Ocean Ecosystems. *Frontiers in Ecology and Evolution* **9** (2021). <https://doi.org/10.3389/fevo.2021.624451>
- 14 Keesstra, S. *et al.* The way forward: Can connectivity be useful to design better measuring and modelling schemes for water and sediment dynamics? *Science of the Total Environment* **644**, 1557-1572 (2018). <https://doi.org/10.1016/j.scitotenv.2018.06.342>

- 15 Whittaker, A. C. How do landscapes record tectonics and climate? *Lithosphere-Ur* **4**, 160-164 (2012). <https://doi.org/10.1130/Rf.L003.1>
- 16 Fryirs, K. (Dis)Connectivity in catchment sediment cascades: a fresh look at the sediment delivery problem. *Earth Surface Processes and Landforms* **38**, 30-46 (2013). <https://doi.org/10.1002/esp.3242>
- 17 Marchi, L., Comiti, F., Crema, S. & Cavalli, M. Channel control works and sediment connectivity in the European Alps. *Sci Total Environ* **668**, 389-399 (2019). <https://doi.org/10.1016/j.scitotenv.2019.02.416>
- 18 Zhang, T. *et al.* Warming-driven erosion and sediment transport in cold regions. *Nature Reviews Earth & Environment* **3**, 832-851 (2022). <https://doi.org/10.1038/s43017-022-00362-0>
- 19 Lane, S. N., Bakker, M., Gabbud, C., Micheletti, N. & Saugy, J.-N. Sediment export, transient landscape response and catchment-scale connectivity following rapid climate warming and Alpine glacier recession. *Geomorphology* **277**, 210-227 (2017). <https://doi.org/10.1016/j.geomorph.2016.02.015>
- 20 Lehner, B., Verdin, K. & Jarvis, A. New Global Hydrography Derived From Spaceborne Elevation Data. *Eos, Transactions American Geophysical Union* **89** (2008). <https://doi.org/10.1029/2008eo100001>
- 21 Lehner, B. & Grill, G. Global river hydrography and network routing: baseline data and new approaches to study the world's large river systems. *Hydrological Processes* **27**, 2171-2186 (2013). <https://doi.org/10.1002/hyp.9740>
- 22 Rigby, R. A. & Stasinopoulos, D. M. Generalized additive models for location, scale and shape. *Journal of the Royal Statistical Society: Series C (Applied Statistics)* **54**, 507-554 (2005). <https://doi.org/10.1111/j.1467-9876.2005.00510.x>
- 23 Chen, S. A., Michaelides, K., Grieve, S. W. D. & Singer, M. B. Aridity is expressed in river topography globally. *Nature* **573**, 573-577 (2019). <https://doi.org/10.1038/s41586-019-1558-8>
- 24 Seybold, H., Berghuijs, W. R., Prancevic, J. P. & Kirchner, J. W. Global dominance of tectonics over climate in shaping river longitudinal profiles. *Nature Geoscience* **14**, 503-507 (2021). <https://doi.org/10.1038/s41561-021-00720-5>
- 25 Pagani, M. *et al.* The 2018 version of the Global Earthquake Model: Hazard component. *Earthquake Spectra* **36**, 226-251 (2020). <https://doi.org/10.1177/8755293020931866>
- 26 Fick, S. E. & Hijmans, R. J. WorldClim 2: new 1-km spatial resolution climate surfaces for global land areas. *International Journal of Climatology* **37**, 4302-4315 (2017). <https://doi.org/10.1002/joc.5086>
- 27 Schwarz, G. Estimating the Dimension of a Model. *The Annals of Statistics* **6**, 461-464 (1978). <http://www.jstor.org/stable/2958889>
- 28 Borselli, L., Cassi, P. & Torri, D. Prolegomena to sediment and flow connectivity in the landscape: A GIS and field numerical assessment. *Catena* **75**, 268-277 (2008). <https://doi.org/10.1016/j.catena.2008.07.006>
- 29 Zanandrea, F., Michel, G. P. & Kobiyama, M. Impedance influence on the index of sediment connectivity in a forested mountainous catchment. *Geomorphology* **351** (2020). <https://doi.org/10.1016/j.geomorph.2019.106962>
- 30 Baartman, J. E. M. *et al.* What do models tell us about water and sediment connectivity? *Geomorphology* **367** (2020). <https://doi.org/10.1016/j.geomorph.2020.107300>

- 31 Perron, J. T. Climate and the Pace of Erosional Landscape Evolution. *Annu Rev Earth Pl*
Sc **45**, 561-591 (2017). <https://doi.org/10.1146/annurev-earth-060614-105405>
- 32 Trauerstein, M., Norton, K. P., Preusser, F. & Schlunegger, F. Climatic imprint on
landscape morphology in the western escarpment of the Andes. *Geomorphology* **194**, 76-
83 (2013). <https://doi.org/10.1016/j.geomorph.2013.04.015>
- 33 Allen, C. D. & Breshears, D. D. Drought-induced shift of a forest-woodland ecotone: Rapid
landscape response to climate variation. *P Natl Acad Sci USA* **95**, 14839-14842 (1998).
<https://doi.org/10.1073/pnas.95.25.14839>
- 34 East, A. E. & Sankey, J. B. Geomorphic and Sedimentary Effects of Modern Climate
Change: Current and Anticipated Future Conditions in the Western United States. *Reviews*
of Geophysics **58** (2020). <https://doi.org/10.1029/2019RG000692>
- 35 Camici, S., Brocca, L., Melone, F. & Moramarco, T. Impact of Climate Change on Flood
Frequency Using Different Climate Models and Downscaling Approaches. *Journal of*
Hydrologic Engineering **19**, 04014002 (2014).
[https://doi.org/doi:10.1061/\(ASCE\)HE.1943-5584.0000959](https://doi.org/doi:10.1061/(ASCE)HE.1943-5584.0000959)
- 36 Yetemen, O., Istanbuluoglu, E. & Vivoni, E. R. The implications of geology, soils, and
vegetation on landscape morphology: Inferences from semi-arid basins with complex
vegetation patterns in Central New Mexico, USA. *Geomorphology* **116**, 246-263 (2010).
<https://doi.org/10.1016/j.geomorph.2009.11.026>
- 37 Chadwick, O. A. *et al.* Shaping post-orogenic landscapes by climate and chemical
weathering. *Geology* **41**, 1171-1174 (2013). <https://doi.org/10.1130/G34721.1>
- 38 Eyring, V. *et al.* Overview of the Coupled Model Intercomparison Project Phase 6 (CMIP6)
experimental design and organization. *Geoscientific Model Development* **9**, 1937-1958
(2016). <https://doi.org/10.5194/gmd-9-1937-2016>
- 39 Gay, A., Cerdan, O., Mardhel, V. & Desmet, M. Application of an index of sediment
connectivity in a lowland area. *J Soil Sediment* **16**, 280-293 (2016).
<https://doi.org/10.1007/s11368-015-1235-y>
- 40 Lin, P. R., Pan, M., Wood, E. F., Yamazaki, D. & Allen, G. H. A new vector-based global
river network dataset accounting for variable drainage density. *Sci Data* **8** (2021).
<https://doi.org/10.1038/s41597-021-00819-9>
- 41 Wasko, C., Nathan, R., Stein, L. & O'Shea, D. Evidence of shorter more extreme rainfalls
and increased flood variability under climate change. *Journal of Hydrology* **603** (2021).
<https://doi.org/10.1016/j.jhydrol.2021.126994>

REVIEWERS' COMMENTS

Reviewer #1 (Remarks to the Author):

No comments.

Reviewer #2 (Remarks to the Author):

I have read the rebuttal letter, focusing on the response to my comments, and the revised manuscript.

My comments regarding the definition(s) of structural/functional connectivity and a discussion of their implications on what the present study observes have been dealt with diligently and satisfactorily. The methods are reproducible to a higher degree now, and options for improving the results are at least mentioned. While I still not fully agree with every single statement, e.g. regarding the nature of the observed correlations between precipitation and IC, it is totally acceptable that the authors' take is a different one; now, the reader can judge better for themselves as some notes of caution have been added.

In my view, this manuscript has matured a lot and hence I recommend acceptance.

Reviewer #1 (Remarks to the Author):

No comments.

Reviewer #2 (Remarks to the Author):

I have read the rebuttal letter, focusing on the response to my comments, and the revised manuscript. My comments regarding the definition(s) of structural/functional connectivity and a discussion of their implications on what the present study observes have been dealt with diligently and satisfactorily. The methods are reproducible to a higher degree now, and options for improving the results are at least mentioned. While I still not fully agree with every single statement, e.g. regarding the nature of the observed correlations between precipitation and IC, it is totally acceptable that the authors' take is a different one; now, the reader can judge better for themselves as some notes of caution have been added. In my view, this manuscript has matured a lot and hence I recommend acceptance.

R: Thank you for the suggestions and feedback on this manuscript, which has helped to greatly improve its overall quality. Below we address the additional comments provided.

My additional reading of and thinking about this study made me add some comments here:

- **Lines 74-76: add “functional” to “will amplify connectivity”**

R: This sentence was supposed to be in relation structural connectivity. However, the “We hypothesize” statement, we originally had written, does seem to lend itself to functional connectivity as you point out. For this reason, we omit this statement to avoid confusion.

- **in line 75 – climate change will first and foremost alter the magnitude and frequency of hydrometeorological drivers that directly govern functional connectivity. Should such changes increase structural connectivity, then this could only take place on a much (!) longer time scale; structural changes are expected to be either immediate (in response to an extreme event e.g. a mass movement) or gradual, needing a long time for transition.**

R: As previously stated, this statement has been removed as it lends itself to functional connectivity and adds confusion to the ending of the introduction.

- **Lines 100ff: This is probably in response to my cautionary statement that a correlation of precipitation and IC could be due (or strongly enhanced) by the fact that precipitation is usually higher in mountains, so that a correlation of precipitation and the index (which is only an index) could be just a coincidence and not a correlation based on a factual correlation of rainfall and structural connectivity. I do acknowledge and accept that the authors checked for a correlation of elevation and precipitation and found none. This is probably due to the global scale of their investigation where extremes such as extremely wet tropical areas or extremely dry areas without a correlation could obscure a clear correlation elsewhere (that definitely exists on a regional scale). The ”cause” of the observed correlation could be that I’d expect drainage density to be higher in wetter areas, leading to higher index**

values as the distance to the closest “IC target” becomes smaller on average. Maybe the authors wish to include such an explanation.

R: Thank you for the suggestion. We have added this explanation near this statement.

- **I am still struggling with the authors’ claim that a regression model of IC (=the proxy of structural connectivity) on climatic proxies (such as today’s annual rainfall, evapotranspiration etc) is applicable to predict future connectivity (using modelled scenarios of the climatic proxies). I have already stated that changes to structural connectivity, except for immediate reactions to some extreme event that re-arranges/re-shapes parts of the landscape, in reaction to climate change could only happen with a long temporal offset as the “new” landscape adapts to a new equilibrium with the changed climate. What I have not written yet (but is equally important):**

- **it is by no means clear that today’s landscape IS in equilibrium with today’s climate (and that this would be the case everywhere) – which is somehow an assumption the authors make when setting up a regression model based on today’s climate and structural connectivity. I’m not claiming that this is not the case, but I feel the authors should be explicit about that important assumption. The judgement whether the regression model and the conclusions are really valid should be done by the readers; my personal disagreement should not lead to a rejection of the manuscript which conveys the authors’ conclusions.**

R: Thank you for the comment and for your willingness to let the reader decide whether our conclusions are valid, even though different from yours. We have added this assumption to lines 191-194 within the manuscript. We urge the readers to consider this when interpreting our projected changes.

- **I am not sure if a change, for example in precipitation, always and everywhere leads to the same topographic adjustment and the associated degree of structural connectivity. For example, it is to be expected that the same change in hydrometeorological forcing has different consequences in areas with different lithology that are otherwise similar or equal. Moreover, landscape evolution might not be determined solely by (changes in) forcing, but also be contingent on “geomorphic history” (area having been glaciated or not, for example) i.e. “path dependence”. This is also a note of caution that I feel the authors should add to the discussion.**

R: To address this comment, we have added a note of caution in lines 210-213.

- **Lines 107-109 (related to the previous comment): Should read, in my opinion: “Finally, it is important to note that we use, for set-up and application of the regression model, climate variables like precipitation and PET over decadal time**

periods; structural connectivity is theorized to be set by landscape evolution that requires much longer temporal scales.” The authors are asked to check whether this somewhat stronger/more explicit wording better conveys the “temporal scale mismatch”.

R: Thank you for the comment. We agree that it is important to convey temporal scale mismatch. However, the supplied revision would add confusion to the manuscript as it discusses a section not yet introduced in the paper. For this reason, we use the main point from the provided statement and modify it accordingly.